# INSTANCE-DEPENDENT EARLY STOPPING

**Suqin Yuan**[1]    **Runqi Lin**[1]    **Lei Feng**[2]*    **Bo Han**[3]    **Tongliang Liu**[1]*
[1] Sydney AI Centre, The University of Sydney
[2] Southeast University [3] Hong Kong Baptist University

## ABSTRACT

In machine learning practice, early stopping has been widely used to regularize models and can save computational costs by halting the training process when the model's performance on a validation set stops improving. However, conventional early stopping applies the same stopping criterion to all instances without considering their individual learning statuses, which leads to redundant computations on instances that are already well-learned. To further improve the efficiency, we propose an Instance-dependent Early Stopping (IES) method that adapts the early stopping mechanism from the entire training set to the instance level, based on the core principle that *once the model has mastered an instance, the training on it should stop*. IES considers an instance as *mastered* if the second-order differences of its loss value remain within a small range around zero. This offers a more consistent measure of an instance's learning status compared with directly using the loss value, and thus allows for a unified threshold to determine when an instance can be excluded from further backpropagation. We show that excluding *mastered* instances from backpropagation can increase the gradient norms, thereby accelerating the decrease of the training loss and speeding up the training process. Extensive experiments on benchmarks demonstrate that IES method can reduce backpropagation instances by 10%-50% while maintaining or even slightly improving the test accuracy and transfer learning performance of a model. Our implementation can be found at https://github.com/tmllab/2025_ICLR_IES.

## 1 INTRODUCTION

Early stopping is a straightforward technique that regulates model training and reduces computational costs by halting the training process when no further improvements are observed in model performance on the validation set (Prechelt, 2002; Raskutti et al., 2014; Caruana et al., 2000; Yuan et al., 2024). Specifically, this method terminates training at the appropriate moment, preventing excessive training while conserving computational resources (Zhang et al., 2021; Belkin et al., 2019; Nakkiran et al., 2021) and reduces the reliance on other computationally intensive regularization methods in model training (Tibshirani, 1996; Hoerl & Kennard, 1970; Goodfellow et al., 2016). The growing size and complexity of models and datasets make these benefits increasingly critical, as they lead to significantly rising computational costs associated with training advanced models (Hestness et al., 2017; Kaplan et al., 2020; Brown et al., 2020; Sorscher et al., 2022; Li et al., 2023b; Gong et al., 2024). In practice, ending training when satisfactory performance is achieved is more practical than pursuing complete convergence, as the cost of complete convergence is excessively high and may not yield evident improvements in performance (Rice et al., 2020; Yang et al., 2020; Sagawa et al., 2020).

Despite the widespread acclaim for the elegance and practicality of the conventional early stopping method, which focuses on the model's performance on the validation set and simultaneously terminates the optimization across the entire training set, this approach lacks flexibility. It does not consider that the model learns different instances at varying rates and stages (Zhang et al., 2021; Arpit et al., 2017; Toneva et al., 2018; Wen et al., 2022). Consequently, this can lead to redundant computations, as the model may continue processing instances that are already well-learned until it finally achieves satisfactory performance across the entire dataset. To further enhance the efficiency of early stopping, we propose the *Instance-dependent Early Stopping* (IES) method, which refines the idea of early stopping from the entire training dataset to the instance level.

---

*Corresponding authors.

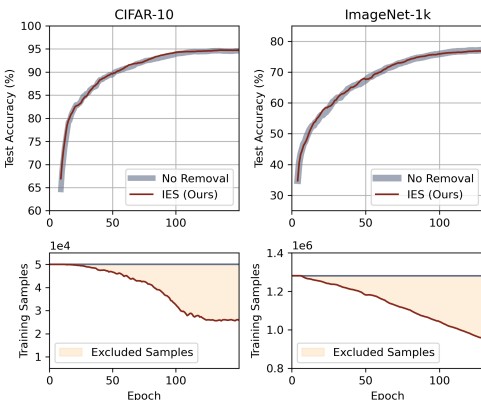

Figure 1: Effectiveness of *Instance-dependent Early Stopping* (IES) on ImageNet-1k and CIFAR-10 datasets. Top row: Test accuracy over the course of training, showing that IES (Ours) achieves comparable accuracy to the baseline (No Removal) despite training on fewer samples. Bottom row: Number of training samples excluded from backpropagation by IES over the course of training. As the model masters more and more samples during the training process, IES allows an increasing number of these *mastered* samples to be excluded from backpropagation, significantly reducing computation while still maintaining the same performance as the baseline method.

The principle of our IES method is simple yet effective: *once the model masters an instance, the training on it should stop*. By enabling the model to dynamically stop the training for individual instances once satisfactory performance is achieved for those specific instances, IES can perform early stopping in a more fine-grained manner. To instantiate the concept of *mastered*, we need a computational efficiency quantitative criterion that can be applied uniformly across all instances. A natural idea is to use the loss value of instances, which has been shown effective for identifying important instances for optimization (Loshchilov & Hutter, 2015; Jiang et al., 2019; Qin et al., 2023). However, due to the differences in optimal loss values across instances arising from factors such as sample complexity (Hacohen & Weinshall, 2019; Wang et al., 2020), inherent ambiguity (Guo et al., 2017; Liang et al., 2017), noise (Zhang et al., 2021; Jiang et al., 2018), and imbalance (Cui et al., 2019; Cao et al., 2019), it may be suboptimal for determining whether an instance has been *mastered*.

In this paper, we propose to use the second-order difference of an instance's loss values $\Delta^2 L_i(w^{(t)})$ across consecutive training epochs as the *mastered* criterion. If, over $k$ epochs, the sum of the absolute values of $\Delta^2 L_i(w^{(t)})$ for an instance $i$ is confined to a small neighborhood around 0, it signifies that the change in the loss tends to be flat and insensitive to parameter updates. Compared with the loss values, the second-order differences of these values for the training data have a lower coefficient of variation in later training stages (Figure 3). This indicates that the second-order difference loss values of *mastered* instances consistently fall within a small range, regardless of the actual loss values of these instances. This consistency allows us to set a unified threshold based on the second-order difference values to determine whether an instance has been *mastered*. Moreover, the proposed criterion is computationally efficient, relying solely on forward propagation.

As shown in Figure 1, as model training progresses and more instances are *mastered*, the IES method allows an adaptive decrease in the number of training instances from backpropagation. This results in significant savings in overall training time and computational costs while obtaining models with comparable performance to the one that is trained using all data. Specifically, the effectiveness of the IES method in accelerating model training progression can be attributed to its ability to allow the model to focus on instances that are not yet *mastered*, which typically have larger gradient norms, thereby speeding up the reduction of the training loss through more effective parameter updates. By effectively identifying and skipping the redundant instances that have already been well-learned and would not significantly contribute to further model performance improvement in the next few epochs, IES achieves comparable results to full-data training. Moreover, by avoiding repeated training on already *mastered* instances, the IES method avoids *Over-Memorization* (Ishida et al., 2020; Zhang et al., 2021; Wen et al., 2024; Lin et al., 2024a;b;c) and enables the model to more rapidly reduce the sharpness of the loss landscape (Dauphin et al., 2014; Foret et al., 2020).

To assess the effectiveness of the IES method, we carried out extensive experiments across various settings. Our findings reveal that IES consistently delivers substantial computational savings in CIFAR and ImageNet-1k tasks, reducing the number of instances that require backpropagation by 10% to 50% without sacrificing model performance. In many cases, IES even slightly enhances the model's generalization performance and improves transferability to downstream tasks. Specifically, fine-tuning models pretrained with IES on ImageNet-1k for the CIFAR and Caltech-101 datasets results in average improvements of 1.5%, compared with models pretrained without IES. Through ablation studies and comparative analysis, we demonstrate that IES outperforms existing samples selection methods and demonstrates robust adaptiveness in hyperparameter selection.

Our main contributions can be summarized as follows:

1. We propose *Instance-dependent Early Stopping* (IES), a method that adaptively stops training at the instance level, allowing for the saving of computational resources while maintaining performance.

2. We introduce a *mastered* criterion based on the second-order differences of sample loss values, providing an unified measure to determine whether a model has fully learned a given instance.

3. We analyze the mechanism behind IES's effectiveness, revealing that it allows the model to focus on instances with larger gradient norms and reduces the sharpness of loss landscape more rapidly.

## 2 RELATED WORK

IES is closely related to multiple active machine learning research areas. We review key studies in these fields, underscoring IES's distinct features.

*Sample Selection* has been widely used to improve the efficiency and robustness of deep learning model training. The main idea is to assign higher probabilities to examples to be trained that are *informative* (Alain et al., 2015; Katharopoulos & Fleuret, 2017; 2018), *unique* (Loshchilov & Hutter, 2015; Chang et al., 2017; Shi et al., 2021) or *confident* (Khim et al., 2020). Related associated distillation and selection algorithms usually incur additional costs. Static selection typically requires preliminary calculations before training or in the early stages of training, with related studies including *Data Pruning* (Toneva et al., 2018; Paul et al., 2021; Killamsetty et al., 2021b) and *Core Set* (Huggins et al., 2016; Huang et al., 2018; Braverman et al., 2022; Xia et al., 2022; 2024), etc., with the goal of finding a small subset from all training data that can represent the entire dataset. Dynamic selection usually involves selecting instances across training process, with related studies including *Dynamic Data Pruning* (Raju et al., 2021; Mindermann et al., 2022; He et al., 2023; Truong et al., 2023; Qin et al., 2023) and *Importance Sampling* (Alain et al., 2015; Katharopoulos & Fleuret, 2017; 2018; Csiba & Richtárik, 2018; Jiang et al., 2019), etc., aimed at focusing training on more informative or confident examples. In the context of deep learning, several methods have been proposed based on different measures of sample "informative", such as gradient norm (Alain et al., 2015; Killamsetty et al., 2021a), loss value (Loshchilov & Hutter, 2015; Schaul et al., 2015; Mindermann et al., 2022), and prediction uncertainty (Chang et al., 2017). Notably, when the gradient of an instance converges to zero, it means that the model's parameters will be insignificant updated based on this particular sample. However, even with efficient gradient computation methods (Wei et al., 2017; Katharopoulos & Fleuret, 2017; 2018), the computational cost of calculating the gradient of each sample based on backpropagation is still high, which hinders the goal of reducing the computational cost of every single run. *Curriculum Learning* (Bengio et al., 2009; Wu et al., 2021; Zhou et al., 2020; Wang et al., 2024b;a; Kumar et al., 2010) is a learning paradigm that aims to improve the efficiency and effectiveness of training by presenting examples in a meaningful order, typically from easy to hard. Several methods have been proposed based on different measures of example difficulty (Weinshall et al., 2018; Saxena et al., 2019; Jiang et al., 2018). IES method can be viewed as a curriculum learning method design for end of training, focusing on the model's mastery of instances.

Although IES and existing sample selection techniques share the common goal of improving training progression via training on a selected subset of training instances, our method distinguishes itself through its focus on whether "the model has already fully learned an instance", i.e., *mastered*. This unique perspective allows IES to adaptively adjust the proportion of instances participating in training at different stages, thereby eliminating the need for pre-set training schedules or removal rates.

## 3 METHODOLOGY

To refine the advantages of early stopping to the instance level, we proposed a simple principle that, *once the model masters an instance, the training on it should stop.* To operationalize this idea, we introduce a criterion for identifying instances that the model has been *mastered*, as detailed in Section 3.1. Building on this foundation, we propose *Instance-dependent Early Stopping* (IES) to promote model training progression, as shown in Section 3.2. Furthermore, we demonstrate the efficiency and effectiveness of the IES method, as discussed in Section 3.3. All toy experiments presented in this section use a standard ResNet-18 backbone trained on the CIFAR-10 dataset. For detailed experiment settings, please refer to Section 4 and Appendix A and B.5.

**Preliminaries.** - The *Hessian matrix*, $H = \nabla^2 L(w)$, characterizes the loss function's curvature by its eigenvalues and eigenvectors: $H = Q\Lambda Q^T = \sum_{i=1}^{n} \lambda_i q_i q_i^T$. $Q$ is an orthogonal matrix of eigenvectors, and $\Lambda$ is a diagonal matrix of eigenvalues $\lambda_i$, describing the curvature in various directions. Higher eigenvalues imply steeper curvatures, complicating optimization (Li et al., 2017).

- $\nabla$ represents the gradient operator, for example, $\nabla L(w)$ represents the gradient of the loss function $L$ at the parameter $w$. $\Delta$ represents the difference operator, $\Delta^2 L_i(w^{(t)})$ represents the second-order difference of the loss function for sample $i$ over three consecutive time steps $t$, $t-1$, and $t-2$.

## 3.1 THE MASTERED CRITERION

Previous studies have shown that different instances contain varying information and have inconsistent impacts on model learning at different training stages (Zhang et al., 2021; Arpit et al., 2017; Toneva et al., 2018; Huang et al., 2022; 2024; Li et al., 2023a; Hong et al., 2024). This suggests that if certain instances have been well-learned by the model early in the training process, their contribution to model performance improvement may diminish or even become redundant as training progresses. To apply the idea of early stopping at the instance level and improve training efficiency, we need a simple and computationally efficient method to assess the model's learning status on each sample and identify these redundant instances that would not significantly contribute to further model performance improvement in the next few epochs, which we refer to as *mastered* instances.

To efficiently identify which and when an instance is *mastered*, we construct a criterion based on the $N$-th order difference of sample loss, which only relies on forward propagation. Intuitively, the loss of a *mastered* instance should be relatively stable. Specifically, when an instance $i$ is well fitted by the current model parameters $w^{(t)}$ or is insensitive to their recent update, the associated loss $L_i(w^{(t)})$ will be small or reach a plateau, and thus the $N$-th order difference of the loss will approach zero. To formalize this, when the $N$-th order difference of the loss for sample $i$ falls beneath a specified small positive threshold $\delta$, sample $i$ is considered to be *mastered* by the model parameters $w$ and state $t$, which can be expressed as:

$$\Delta^N L_i(w^{(t)}) < \delta, N = \{0, 1, 2, ...\}. \quad (1)$$

To demonstrate the effectiveness of the *mastered* criteria, we experimentally tracked the number of instances that meet the *mastered* criteria during the training process. As shown in Figure 2, the *mastered* criteria enable adaptive sample selection throughout the learning process, allowing the model to dynamically adjust the size of the training set participating in backpropagation according to the evolving requirements. During the initial stages of training, the model has scarcely learned any instances, so the *mastered* criteria retain most instances for backpropagation, as almost every sample can provide useful information. As training progresses, the model gradually masters more instances, leading to an adaptive decrease in the number of retained training instances commensurate with the training progress. The *mastered* criteria adaptively remove these fully learned redundant instances from backpropagation, enabling the model to focus on the remaining samples. Compared to methods that dynamically sample a fixed proportion of important instances (Raju et al., 2021; Mindermann et al., 2022; He et al., 2023; Qin et al., 2023), stopping training on *mastered* instances, provides a more adaptive and efficient approach to instance selection based on the model's learning progress.

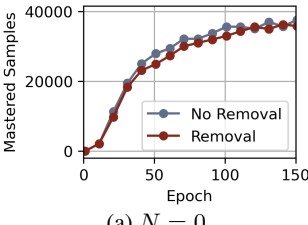 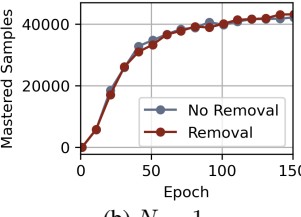 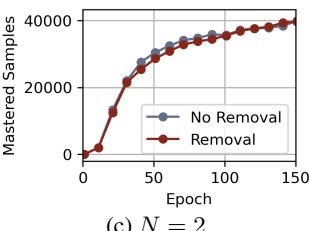

(a) $N = 0$          (b) $N = 1$          (c) $N = 2$

Figure 2: The curves show the number of instances that meet the corresponding mastered criteria (N = {0, 1, 2}, $\delta = 1e^{-4}$) as the training epochs progress, under two scenarios: excluding the mastered instances from backpropagation and allowing the mastered instances to participate in backpropagation. The proximity of the curves suggests that the model can maintain its "mastered" on the mastered instances without the need for actively repeated training on them.

It is noteworthy that the number of the model *mastered* instances remains nearly the same regardless of whether the instances satisfying the *mastered* criteria continue to participate in backpropagation or not, as shown in Figure 2. This observation, which is particularly evident for $N = 1$ and $N = 2$, suggests that the model can maintain its learned state on the *mastered* instances even without repeatedly training on them. The *mastered* criterion thus provides an effective way to identify redundant instances during training, allowing the model to exclude them from backpropagation with minimal impact on model performance on these instances.

## 3.2 INSTANCE-DEPENDENT EARLY STOPPING (IES)

Building upon the *mastered* criterion, we propose the *Instance-dependent Early Stopping* (IES) method, allowing the model to *stop training on an instance once it has been mastered*. Although using the loss value of instances (i.e., $N = 0$) has been widely adopted as a method to identify important instances for current optimization (Loshchilov & Hutter, 2015; Jiang et al., 2019), different instances may have different optimal loss values $L_i(w^*)$ due to factors such as sample complexity (Hacohen & Weinshall, 2019; Wang et al., 2020), noise (Zhang et al., 2021; Jiang et al., 2018), and imbalance (Cui et al., 2019; Cao et al., 2019). This poses a challenge in simply using loss value to construct mastered criterion for IES. If the mastered criterion were to directly depend on the absolute loss value, it would require setting different thresholds for each sample, which can be impractical and expensive in large-scale datasets. In this work, we use the second-order difference to identify the mastered instances, which quantifies the rate of change in the loss for sample $i$ across three consecutive epochs, $t^{th}$, $(t-1)^{th}$, and $(t-2)^{th}$ training epochs. The second-order difference is defined as:

$$\Delta^2 L_i(w^{(t)}) = [L_i(w^{(t)}) - L_i(w^{(t-1)})] - [L_i(w^{(t-1)}) - L_i(w^{(t-2)})]$$
$$= L_i(w^{(t)}) - 2L_i(w^{(t-1)}) + L_i(w^{(t-2)}). \tag{2}$$

By quantifying the rate of change in the loss for each instance around the current parameters $w^{(t)}$, the second-order difference effectively captures the stability of the loss function, regardless of the specific value of $L_i(w^*)$. This property allows for using a unified threshold $\delta$ across all instances, greatly simplifying the implementation and management of the mastered criterion. To further validate this advantage, we conducted experiments as shown in Figure 3. We calculate the zero-order (loss value), first-order, and second-order differences for each sample's loss during training on CIFAR-10. Subsequently, we computed the coefficient of variation (CV) for these differences to represent the degree of dispersion in the data. Experimental results show that when $N = 1, 2, 3$, the CV of their high-order differences of loss value exhibits a trend of first rising and then falling, eventually stabilizing at a lower level. This indicates that in the early stages

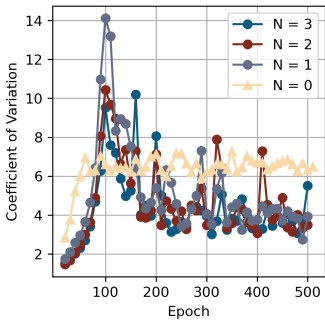

Figure 3: Coefficient of variation (CV) of different orders of loss differences during training.

of training, when only some samples are sufficiently learned, there is significant variability in the higher-order differences of loss value among different samples. As training progresses into the mid-to-late stages, and as most instances become sufficiently learned, all instances exhibit more similar values in their higher-order differences of loss value. Therefore, we can use a fixed threshold to uniformly determine whether an instance has been *mastered*. Further experiments confirm that the high-order difference has relatively smaller CV values, please refer to Appendix C. In our subsequent experiments (Section 4.2), we further evaluate the IES method under different criteria.

Accordingly, based on the above analysis and experimental validation, an instance $i$ is considered *mastered* when the cumulative magnitude of these second-order differences of loss value falls beneath a specified small positive threshold $\delta$, which is formally expressed as:

$$\left| \Delta^2 L_i(w^{(t)}) \right| < \delta. \tag{3}$$

Ultimately, IES consists of two key stages: filtering out mastered instances in the full training-set $\mathcal{D}^{(0)}$ through forward propagation, and removing mastered instances and only optimizing not-yet mastered instances $\mathcal{D}^{(t)}$ through backpropagation, as detailed in Algorithm 1.

---

**Algorithm 1** Instance-dependent Early Stopping (IES)

---

**Require:** Full training-set $\mathcal{D}^{(0)}$, validation-set $\mathcal{V}$, model $f_\theta$, threshold $\delta$, max epochs $T$
 1: Initialize model parameters $w^{(0)}$
 2: **for** $t = 1$ to $T$ **do**
 3:     Forward pass on model $f$ to compute loss $L_i(w^{(t)})$ for each sample $i \in \mathcal{D}^{(0)}$
 4:     Calculate second-order differences: $\Delta^2 L_i(w^{(t)}) = L_i(w^{(t)}) - 2L_i(w^{(t-1)}) + L_i(w^{(t-2)})$
 5:     Identify *mastered* instances: $\mathcal{M}^{(t)} = \left\{ i \in \mathcal{D}^{(0)} : \left| \Delta^2 L_i(w^{(t')}) \right| < \delta \right\}$
 6:     Update dataset for next epoch: $\mathcal{D}^{(t)} = \mathcal{D}^{(0)} \setminus \mathcal{M}^{(t)}$
 7:     **if** $\mathcal{D}^{(t)}$ is empty **or** conventional early stopping criterion$(\mathcal{V}, w^{(t)})$ **then**
 8:         Break {Stop if all instances *mastered* or conventional early stopping triggered}
 9:     **end if**
10:     Update model parameters $w^{(t)}$ using instances in $\mathcal{D}^{(t)}$
11: **end for**

---

### 3.3 INSTANCE-DEPENDENT STOPPING TO ACCELERATE MODEL TRAINING PROGRESSION

The proposed Instance-dependent Early Stopping (IES) method significantly reduces computational costs, as shown in its twofold impact: (1) IES achieves comparable performance to the baseline while requiring fewer backpropagation instances, and (2) IES surpasses the baseline's performance with the same amount of backpropagation. This subsection presents experimental results and analysis to showcase the IES's effectiveness and effective in accelerating model training progression. A detailed analysis demonstrating that IES does not lead to catastrophic forgetting is provided in Appendix H.

**Less backpropagation, similar performance.** Our proposed method achieve comparable performance to the baseline method while using fewer instances in backpropagation. As detailed in Section 3.1, the effectiveness of using fewer instances in backpropagation without compromising performance is achieved through the precise identification of mastered instances. As shown in Figure 4, our method reduces the number of training instances in backpropagation by approximately 40%, resulting in a savings of nearly 30% in total computational cost while maintaining generalization performance.

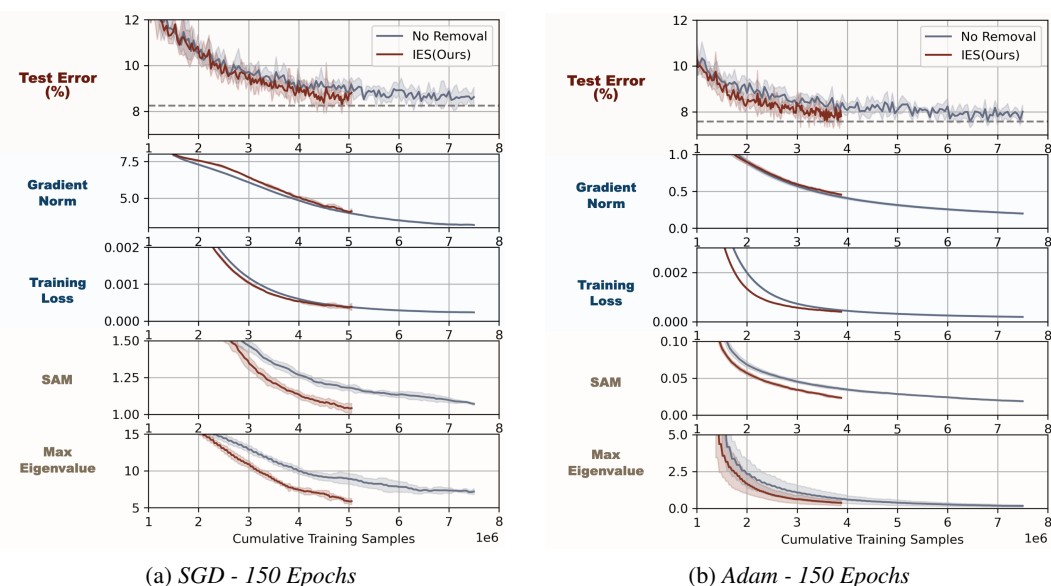

(a) *SGD - 150 Epochs*          (b) *Adam - 150 Epochs*

Figure 4: Comparison of model performance metrics between the IES method and the baseline method over the same number of backpropagation training instances. The metrics include test error, gradient norm, training loss, sharpness-aware minimization (SAM) value, and the maximum eigenvalue of the Hessian matrix. IES consistently outperforms the baseline in test error and reduces training loss, SAM value, and the maximum eigenvalue more effectively, indicating a faster progression in model training. We use ResNet-18 on the CIFAR-10 dataset in this experiment. Further detailed experimental settings can be found in Appendix B and Section 4.

**Same backpropagation, better performance.** Our proposed method consistently achieves better performance than the baseline method at the same number of backpropagation instances. To further demonstrate the superiority of our method, we conduct analysis from following experiments.

*Larger Gradient Norms.* As shown in Figure 4, the average mini-batch gradient norm for instances selected by the IES method is consistently higher than that of the baseline method, which uses all training data. By typically selecting instances with larger gradient norms for backpropagation, the IES method tends to make parameter updates more effective at reducing training loss.

*Faster Reduce Sharpness.* We conducted experiments to investigate the changes of model's sharpness during the training process. Sharpness (Hochreiter & Schmidhuber, 1997; Andriushchenko & Flammarion, 2022; Huang et al., 2023) often refers to the steepness of the loss function near the solution and is closely related to the large eigenvalues of the Hessian matrix (Dinh et al., 2017; Tsuzuku et al., 2020). We compared the changes in the largest eigenvalue of the Hessian matrix and the Sharpness-Aware Minimization (SAM) value (Foret et al., 2020). As shown in Figure 4, IES reduces the largest eigenvalue more quickly and consistently achieves lower SAM values compared with the baseline method. The faster reduction in the largest eigenvalue suggests that IES can more targetedly reduce steepness in these sharp directions of the loss landscape, thereby reducing the overall "sharpnes" more quickly. A lower SAM value indicates a flatter minima with lower sharpness. These empirical observations together support that IES can more targetedly reduce steepness in these sharp directions of the loss landscape (Li et al., 2017; Keskar et al., 2016; Neyshabur et al., 2017; Dauphin et al., 2014), thereby reducing the overall sharpness more quickly.

## 4 EXPERIMENTS

In this section, we empirically demonstrate the effectiveness of *Instance-dependent Early Stopping* method. In Section 4.1, we validate the broad applicability of our proposed method across different settings. Furthermore, we demonstrate through experiments that applying our proposed method can improve the transferability of models. In Section 4.2, we compare our proposed IES method with other methods and different instance-level stopping criteria. We showcase the capability of our method to maintain model performance across a wide range of hyperparameters. Section 4.3 demonstrates our method's applicability for more efficient training and its use in high-level tasks such as segmentation and detection. The empirical evidence indicates that our proposed Instance-dependent Early Stopping method can effectively reduce computational overhead under various settings, outperforming existing baselines while simultaneously enhancing the transfer learning capabilities of models.

### 4.1 EFFECTIVENESS OF IES

To evaluate the effectiveness of our proposed IES method, we conducted extensive experiments under various settings, including different datasets, network architectures, and optimizers. Table 1 and 2 demonstrate the consistent performance of the IES method across these settings. It is worth noting that IES achieves lossless acceleration for model training; if a $1\%$ generalization performance decrease is acceptable, even more substantial acceleration can be obtained, as detailed in Section 4.2.

Our evaluations confirmed the effectiveness of the IES method across multiple datasets. These datasets comprise *CIFAR-10*, *CIFAR-100* (Krizhevsky et al., 2009), and *ImageNet-1k* (Deng et al., 2009). For the *CIFAR* and the *ImageNet-1k* tasks, we train for 200 and 150 epochs, respectively. For *ImageNet-1k* task, we follow Qin et al. (2023) and anneal in the last 10% of epochs. We employ different optimizers such as *SGD* with *Momentum* (Robbins & Monro, 1951; Polyak, 1964), *Adam* (Kingma, 2014), and *AdamW* (Loshchilov & Hutter, 2017) to demonstrate that the IES method remains resilient to reasonable variations across multiple optimizers and learning rate schedulers.

We use different SGD learning rate scheduler settings: *SGD(F)*, *SGD(L)*, *SGD(M)*, and *SGD(E)*, which represent SGD with a fixed learning rate, a linearly decaying learning rate, a multi-step decaying learning rate, and an exponentially decaying learning rate scheduler, respectively. For *CIFAR*, we set base $\delta = 1e^{-3}$; and for *ImageNet-1k*, we set $\delta = 1$. Further, we verified the effectiveness of our proposed IES method over several commonly used deep learning models, including *ResNet* (He et al., 2016), *VGG* (Simonyan & Zisserman, 2014), and *DenseNet* (Huang et al., 2017). More detailed experimental settings and additional results can be found in Appendix B.

Table 1: Effectiveness of IES-2nd across various settings. (5 runs, mean±std)

| | CIFAR-10 | | | CIFAR-100 | | |
|---|---|---|---|---|---|---|
| *Architectures* | ResNet-18 | ResNet-50 | VGG-16 | ResNet-34 | ResNet-101 | DenseNet-121 |
| No Removal | 92.9%±0.1% | 93.3%±0.1% | 90.9%±0.2% | 69.8%±0.4% | 71.9%±0.5% | 73.4%±0.0% |
| IES (Ours) | 92.9%±0.2% | 93.1%±0.1% | 90.7%±0.2% | 69.6%±0.3% | 72.2%±0.5% | 73.3%±0.2% |
| Mini-batch Saved | 54.6% | 48.5% | 30.4% | 29.9% | 28.3% | 33.4% |
| *Optimizers* | SGD(F) | SGD(L) | AdamW | SGD(F) | SGD(E) | AdamW |
| No Removal | 92.1%±0.1% | 95.2%±0.1% | 92.6%±0.1% | 71.4%±0.5% | 77.6%±0.4% | 69.6%±0.3% |
| IES (Ours) | 92.4%±0.0% | 95.1%±0.1% | 92.7%±0.1% | 72.3%±0.4% | 77.4%±0.4% | 69.7%±0.5% |
| Mini-batch Saved | 37.3% | 26.4% | 47.2% | 9.3% | 27.3% | 17.4% |
| Avg. Mini-batch Saved | **40.7%** | | | **24.3%** | | |
| Avg. Wall-time Speedup | $\sim 1.4\times$ | | | $\sim 1.2\times$ | | |

Table 2: Effectiveness of IES-2nd in ImageNet-1k task. (1 run)

| | DenseNet-121 | ResNet-34 | | ResNet-101 |
|---|---|---|---|---|
| Methods | AdamW | AdamW | SGD(M) | SGD(E) |
| No Removal | 69.0% | 68.0% | 74.1% | 77.4% |
| IES (Ours) | 68.8% | 68.0% | 74.3% | 77.4% |
| Mini-batch Saved | 31.6% | 28.7% | 30.3% | 34.2% |
| Avg. Wall-time Speedup | $\sim 1.3\times$ | | | |

Table 3: Transfer performance of IES-2nd pretrained model on ImageNet-1k task. (5 runs, mean±std)

| | ResNet-101 | | DenseNet-121 | |
|---|---|---|---|---|
| Transfer Tasks | IES (Ours) | No Removal | IES (Ours) | No Removal |
| *ImageNet-1k –> CIFAR-10* | **81.2%±0.1%** | 80.3%±0.2% | **78.6% ± 0.2%** | 77.3% ± 0.2% |
| *ImageNet-1k –> CIFAR-100* | **57.5%±0.2%** | 55.6%±0.2% | **53.0% ± 0.2%** | 52.3% ± 0.2% |
| *ImageNet-1k –> Caltech-101* | **59.9%±0.8%** | 57.4%±1.2% | **50.9% ± 1.6%** | 49.5% ± 1.5% |

To quantify the computational resources saved by the IES method, we consider two following metrics:

- *Mini-batch Saved.* We calculate the percentage of mini-batch saved. This metric directly reflects the reduction in the number of instances of backpropagation computations.

- *Wall-time Speedup.* We measure the training time speedup achieved by the IES method compared with full data training. This metric provides a realistic assessment of the time savings. We report the average training speedup on CIFAR-10, CIFAR-100, and ImageNet-1k tasks.

The IES method primarily saves computational resources by reducing the backpropagation steps, which constitute the most time-consuming part of the training process. However, the forward pass still needs to be computed for all instances to obtain their predictions and determine which instances should be stopped based on the *mastered* criterion. The experimental results demonstrate that the IES method can save 10% to 55% of mini-batch computations and speedup 20% to 40% of training time, while maintaining test accuracy comparable to full data training. The empirical evidence indicates the effectiveness of the IES method in reducing computational costs without compromising performance.

**Transferability.** To further evaluate the effectiveness of the IES method, we investigated its impact on the transfer learning of models. We first pretrained models on the ImageNet-1k dataset using IES and the baseline method without instance stopping. Then, we fine-tuned only the classification head of these pretrained models using the model from the last epoch of pretraining on several downstream tasks, including CIFAR-10, CIFAR-100, and Caltech-101 (Li et al., 2022) datasets. As shown in Table 3, models pretrained with IES consistently outperform those pretrained without instance removal across all transfer learning tasks while achieving the comparable test accuracy on the ImageNet-1k. After fine-tuning for 1 epoch, the IES pretrained model surpassing the baseline by 0.9%, 1.9%, and 2.5% on CIFAR-10, CIFAR-100, and Caltech-101, respectively. Similar improvements are observed on DenseNet-121 and on more epoch fine-tuning, as shown in Table 3 and 7 in Appendix B.4.

These results align with the discussion in Section 3.3, suggesting that IES can more effectively reduce the sharpness of the loss landscape during pretraining. By instance-dependent stopping of instance training, IES saves computational resources while potentially contributing to a more favorable loss landscape and more transferable performance. Consequently, models pretrained with IES exhibit better transfer learning performance, adapting more effectively to new tasks with limited fine-tuning.

## 4.2 EFFICIENCY OF IES

Building upon the IES method's demonstrated ability to accelerate training without performance loss, this section further explores its efficiency. We compare IES with different sample selection criteria, analyze the impact of different $\delta$ value settings on performance, and investigate the potential for additional acceleration when allowing a slight decrease in performance.

**Comparison with other sample selection methods.** We compare our proposed IES method with *Random Remove* and *Small Loss & Rescale* (Qin et al., 2023), under different Total Excluded Samples values. Total Excluded Samples values represent the proportion of samples removed from the backpropagation during training. *Random Remove* method randomly removes a certain proportion of samples from backpropagation in each training epoch, while *Small Loss & Rescale* randomly prunes samples with smaller loss values and amplifies the gradients of the remaining small-loss samples. As shown in Figure 5, experimental results on both CIFAR-10 and CIFAR-100 datasets demonstrate that the *Random Remove* method significantly reduces model performance. Although the *Small Loss & Rescale* method improve results, its performance still falls behind IES. Moreover, among different IES configurations, using second-order differences outperforms other configurations in most cases, which aligns with our analysis in Section 3.2. Further details are in Appendix D.

**Analysis of setting $\delta$ values.** To further evaluate the robustness of the IES method, we expanded upon the previous comparison by setting a broader range of $\delta$ values, observing their impact on sample exclusion and model accuracy. As shown in Figure 5 (lower row), we varied the $\delta$ value used in the IES method by multiplying the selected $\delta$ value (set to 0.001) by scales of $\{0.01, 0.1, 1, 10, 100\}$. Notably, even as $\delta$ varied across four orders of magnitude, the IES method maintained the test accuracy within approximately 2% of the baseline performance. This highlights the significant stability and adaptability of the IES method across a wide range of $\delta$ settings, enabling its effective implementation in diverse scenarios without the need for precise fine-tuning of the $\delta$ parameter.

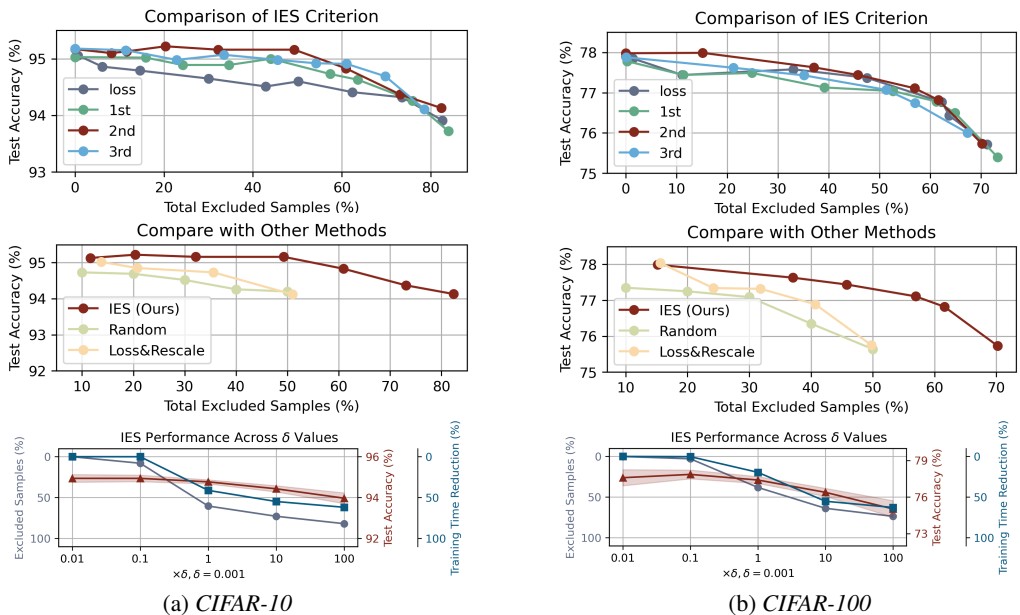

(a) *CIFAR-10*                    (b) *CIFAR-100*

Figure 5: Comparison of the proposed IES method of different IES criteria (loss, 1st, 2nd, and 3rd order differences) with other sample selection methods under different Total Excluded Samples values on both CIFAR datasets. The lower subfigure illustrates the effect of varying $\delta$ values used in IES methods on training time reduction, sample removal, and model performance (3 runs, mean±std).

Table 4: Comparison of IES and other data efficiency methods. (3 runs, mean±std)

| Computation Speedup | Methods | CIFAR-10 | CIFAR-100 |
|---|---|---|---|
| 1.0× | Baseline (No Removal) | 94.3%±0.3% | 77.0%±0.4% |
| ~ 2.0× | Conventional Early Stopping | 90.4%±0.5% | 68.7%±0.5% |
| | SB (Jiang et al., 2019) | 93.0%±0.1% | 70.6%±0.5% |
| | DIHCL (Zhou et al., 2020) | 93.4%±0.2% | 74.3%±0.2% |
| | EfficientTrain (Wang et al., 2024b) | 91.5%±0.2% | **75.0%**±**0.1%** |
| | IES (Ours) | **93.7%**±**0.4%** | 74.9%±0.5% |

## 4.3 FURTHER ANALYSIS

This section explores the scalability of IES for further acceleration, focusing on: (1) accelerating training while tolerating minor performance loss, and (2) maintaining accuracy while achieving targeted training speedups. Additionally, we evaluate the efficacy of IES for high-level vision tasks. Furthermore, we demonstrate the robustness of our proposed IES method under the common scenario of learning with noisy labels (Xia et al., 2020a), as detailed in Appendix G.

**Tolerating 1% performance loss.** While IES aims to maintain test accuracy compared to full data training, it also has potential for further acceleration if a slight decrease in test accuracy is acceptable. By allowing a 1% reduction in test accuracy, we observed that IES can achieve even greater computational savings. For the ImageNet-1k dataset, IES can save up to 40% of backpropagation. As shown in Figure 5 (upper row), for the CIFAR-10 and CIFAR-100 datasets, IES can save up to 80% and 60% of backpropagation, respectively. These results demonstrate that IES can be flexibly adjusted to prioritize either improving efficiency without performance loss (a "free lunch") or further accelerating training within an acceptable range of performance degradation.

**Achieving 2.0× training speedups.** To further evaluate the efficacy of our proposed IES method in scenarios prioritizing efficient training, we conducted a comparison with several data efficiency methods: conventional early stopping, importance sampling (Jiang et al., 2019) *SB*, hardness-based curriculum learning (Zhou et al., 2020) *DIHCL*, and resizing-based curriculum learning (Wang et al., 2024b) *EfficientTrain* methods. For a fair comparison, we set the target computational acceleration to approximately 2.0 times across all methods. The detailed settings and 3.0 times acceleration results are provided in Appendix F. As shown in Table 4, these comparisons further demonstrate that IES performs effectively in accelerating model training while maintaining model performance.

**High-level vision tasks.** To further validate the applicability of the IES method, we conducted experiments on two high-level tasks: object detection and semantic segmentation. Specifically, we integrated our proposed IES method into the baseline methods Faster R-CNN (Ren, 2015) and DeepLab v3 (Chen, 2017), respectively. For both task, we use the PASCAL VOC datasets (Everingham et al., a;b). A brief

| | Faster R-CNN (*mAP*) | DeepLab v3 (*mIoU*) |
|---|---|---|
| No Removal | 70.2%±0.2% | 76.2%±0.2% |
| IES (Ours) | 70.2%±0.1% | 76.1%±0.2% |
| Mini-batch Saved | 20.0% | 14.0% |

Table 5: Effectiveness of the IES on object detection and segmentation model training tasks. (3 runs, mean±std)

overview of the results of model training is reported in the Table 5. Further details are in Appendix E.

## 5 CONCLUSION

In this work, we propose an *Instance-dependent Early Stopping* (IES) method that adapts the early stopping mechanism from the entire training set to the instance level. IES considers an instance as *mastered* if the second-order differences of its loss value remain within a small range around zero, allowing for a unified threshold to determine when an instance can be excluded from further backpropagation. Extensive experiments demonstrate the effectiveness of IES in reducing computational cost while maintaining model performance and transferability.

**Limitation.** While the choice of using the second-order difference as the removal criterion for IES has been validated through experiments, a comprehensive theoretical analysis of its superiority remains an open research question. The impact of the IES method on fairness is evaluated in Appendix I, but it still has not been thoroughly investigated.

ACKNOWLEDGMENTS

The authors would like to thank the anonymous reviewers for their insightful and constructive comments. Suqin Yuan extends special thanks to Muyang Li, Runnan Chen, Xiu-Chuan Li, Jun Wang, Li He, and Yuhao Wu for their valuable advice and computing support. Tongliang Liu is partially supported by the following Australian Research Council projects: FT220100318, DP220102121, LP220100527, LP220200949, IC190100031. BH was supported by RGC Young Collaborative Research Grant No. C2005-24Y, NSFC General Program No. 62376235, and Guangdong Basic and Applied Basic Research Foundation Nos. 2022A1515011652 and 2024A1515012399. This research was undertaken with the assistance of resources from the National Computational Infrastructure (NCI Australia), an NCRIS enabled capability supported by the Australian Government. Suqin Yuan is partially supported by the OpenAI Researcher Access Program.

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

## A  QUICK START GUIDE FOR EXPERIMENTAL SETUP.

**Framework**: `PyTorch, Version 1.11.0.`

**Architecture**

- **Model Type**: Standard ResNet-18 for CIFAR-10, ResNet-34 for CIFAR-100, and ResNet-101 for ImageNet-1k. We do not incorporate dropout.

**Parameters**

- **Batch Size**: {64} for CIFAR and {128} for ImageNet-1k.
- **Training Epochs**: 200 epochs for CIFAR. 150 epochs for ImageNet.
- **Loss Function**: Utilizes the `CrossEntropyLoss` from the `nn` module.

**Dataset & Pre-processing**

- **Normalization**: We employ the `torchvision.transforms` module to adjust pixel values across all images, ensuring they scale uniformly within the 0 to 1 range.
- **Cropping**: We implement a random cropping strategy. Initially, optional padding is applied to each 32x32 image, from which we then extract random 32x32 crops.
- **Rotation**: The images are subject to random rotations with an allowable variation up to ±15 degrees to enhance model robustness against orientation changes.
- **Label Smoothing**: Label smoothing is not incorporated in our pipeline.

## B  DETAILS OF EXPERIMENTS

We provide comprehensive details on the experiments conducted to validate the effectiveness of the Instance-dependent Early Stopping (IES) method. The main results are presented in Section 4, Table 1 and Table 2. Here, we elaborate on the experimental setup across various configurations, covering a wide range of settings typically employed in training deep learning models, including different network architectures, datasets, hyperparameters, and optimizers. Unless otherwise specified, the parameters and components remains consistent with the base model in Appendix A.

### B.1  NETWORK ARCHITECTURES

#### B.1.1  RESNET (HE ET AL., 2016)

- **Variants**: *ResNet-18, ResNet-34, ResNet-50, ResNet-101*.
- **Implementation**:
  - ResNet-18 and ResNet-50 for *CIFAR-10*.
  - ResNet-34 and ResNet-101 for *CIFAR-100*.
  - ResNet-34 and ResNet-101 for *ImageNet-1k*.

#### B.1.2  VGG-16 (SIMONYAN & ZISSERMAN, 2014)

- **Implementation**:
  - Used for *CIFAR-10*.

#### B.1.3  DENSENET-121 (HUANG ET AL., 2017)

- **Implementation**:
  - Used for *CIFAR-100* and *ImageNet-1k*.

## B.2    DATASETS

### B.2.1    CIFAR-10 AND CIFAR-100 (KRIZHEVSKY ET AL., 2009)

- **Description**: 10 classes (*CIFAR-10*) and 100 classes (*CIFAR-100*), 50,000 training and 10,000 test images each.
- **Preprocessing**: Normalization (mean and std), random cropping, horizontal flipping.

### B.2.2    IMAGENET-1K (DENG ET AL., 2009)

- **Description**: 1,000 classes, over 1 million labeled images.
- **Preprocessing**: Normalization (mean and std), random cropping, horizontal flipping.

### B.2.3    CALTECH-101 (LI ET AL., 2022)

- **Description**: 101 object categories
- **Preprocessing**: Normalization (mean and std), random cropping, horizontal flipping.

## B.3    HYPERPARAMETERS AND OPTIMIZATION

- **Batch Sizes**: 64 for *CIFAR* and *Caltech-101*, and 128 for *ImageNet-1k*.
- **$\delta$ settings**: base $\delta = 1e^{-3}$ for *CIFAR*, and base $\delta = 1$ for *ImageNet-1k*.
- **Optimizer settings**: For SGD, momentum=0.9, weight_decay=5e-4.
  - SGD(F) - lr = 0.001.
  - SGD(L) - lr = 0.1, scheduler:
    `LinearLR(_,start_factor=1,end_factor=0.01,total_iters=150)`.
  - SGD(M) - lr = 0.1, scheduler:
    `MultiStepLR(_, milestones=[50, 100], gamma=0.1)`.
  - SGD(E) - lr = 0.1, scheduler: `ExponentialLR(_, gamma=0.96)`.
  - Adam (Kingma, 2014) - lr = 0.001.
  - AdamW (Loshchilov & Hutter, 2017) - lr = 0.001, weight_decay=0.01.
- **Annealing** (Qin et al., 2023): For the *ImageNet-1k* task, we switch to using the full training data for the last 10% of the training epochs to give better stability.

## B.4    TRANSFER LEARNING EXPERIMENTS

**Fine-tuning Setup**:

We selected the model checkpoints at the 100th epoch for ResNet-101/AdamW and DenseNet-121/AdamW follow settings from Table 2 experiments. The models were fine-tuned using both the IES method and full-data training. During fine-tuning, only the classification head of the models is updated, while the rest of the model parameters were frozen.

**Experimental Setup**:

The main experimental settings were consistent with those described in Appendix A. The models were fine-tuned for 1 or 5 of epochs using the Adam optimizer with a learning rate of 0.001. Notably, data augmentation techniques such as cropping and rotation were not applied, and all images were resized to a fixed resolution of 224x224. For the Caltech101 dataset, an additional preprocessing step is performed to convert grayscale images to RGB format.

Results for 1-epoch and 5-epoch fine-tuning are displayed in Table 6 and Table 7, respectively.

Table 6: transferability of IES-2nd Pretrained in ImageNet-1k. Fine-tuning 1 epoch. (mean±std)

| Transfer Task | *ResNet-101* | | *DenseNet-121* | |
| --- | --- | --- | --- | --- |
| | IES (Ours) | No Removal | IES (Ours) | No Removal |
| *ImageNet-1k –> CIFAR-10* | **81.2%±0.1%** | 80.3%±0.2% | **78.6% ± 0.2%** | 77.3% ± 0.2% |
| *ImageNet-1k –> CIFAR-100* | **57.5%±0.2%** | 55.6%±0.2% | **53.0% ± 0.2%** | 52.3% ± 0.2% |
| *ImageNet-1k –> Caltech-101* | **59.9%±0.8%** | 57.4%±1.2% | **50.9% ± 1.6%** | 49.5% ± 1.5% |

Table 7: transferability of IES-2nd Pretrained in ImageNet-1k. Fine-tuning 5 epochs. (mean±std)

| | DenseNet-121 | | ResNet-101 | |
|---|---|---|---|---|
| Transfer Task | IES (Ours) | No Removal | IES (Ours) | No Removal |
| *ImageNet-1k –> CIFAR-10* | **82.6%±0.1%** | 81.7%±0.1% | **85.6% ± 0.1%** | 84.6% ± 0.1% |
| *ImageNet-1k –> CIFAR-100* | **61.6%±0.2%** | 60.8%±0.2% | **66.0% ± 0.1%** | 64.4% ± 0.2% |
| *ImageNet-1k –> Caltech-101* | **91.2%±0.2%** | 90.6%±0.3% | **92.7% ± 0.2%** | 92.5% ± 0.3% |

## B.5 EXPERIMENTS IN FIGURE 4

**Setup**:

○ The main experimental settings were consistent with those described in Appendix A.
○ 5 runs, mean±std.
○ Batch size: The batch size is 128.
○ Number of epochs: The models are trained for 150 epochs.
○ $\delta = 1e^{-4}$.

**Evaluation Metrics**:

○ **SAM (Sharpness-Aware Minimization):**
The SAM value is defined as the difference between the perturbed loss and the original loss Foret et al. (2020). The important hyperparameter is rho, which represents the magnitude of the perturbation. In this work, rho is set to 0.05.

○ **Gradient Norm:**
In the training loop, for each batch, calculates the gradient norm. For each parameter p, its gradient norm is calculated as `p.grad.data.norm(2).item() ** 2`. The total gradient norm is the square root of the sum of squares of all parameter gradient norms. Gradient Norm in Figure is the average of gradient norms for all batches in an epoch.

○ **Maximum Eigenvalue of the Hessian Matrix:**
The maximum eigenvalue is estimated using the power iteration Mises & Pollaczek-Geiringer (1929) method to estimate the largest eigenvalue of the Hessian matrix.
The important hyperparameters include:
- n_iters: The number of iterations for the power iteration method, set to 20.
- epsilon: A small positive number for numerical stability, set to $1e^{-10}$.

○ **Training Loss:** The average cross-entropy loss on the training set.

○ **Test Error:** The percentage of misclassified samples in the test set.

## C    COEFFICIENT OF VARIATION

To further investigate the properties of different orders of loss differences as potential *mastered* criteria, we conducted experiments to compare their coefficient of variation (CV). The CV is a standardized measure of dispersion, calculated as the ratio of the standard deviation to the mean:

$$CV = \frac{\sigma}{\mu},$$

where $\sigma$ is the standard deviation and $\mu$ is the mean of the data. We compute the CV values for the zero-order (loss value), first-order, second-order and third-order differences of each sample's loss during training. A lower CV value indicates that the data points are clustered more closely around the mean, while higher CV suggests greater dispersion. Figure 6 presents the CV values for different orders of loss differences over the course of training when using the Adam optimizer. The results show that the second-order difference and the third-order difference generally maintains lower CV values compared to the zero-order and first-order differences throughout the training process.

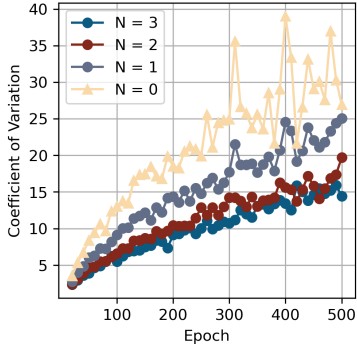

Figure 6: Coefficient of variation (CV) of different orders of loss differences during training. Using Adam optimizer, learning rate = 0.001.

Although the CV values do not converge to a low level in the later stages of training as observed with the SGD optimizer (results on SGD shown in Figure 3), the second-order difference and the third-order difference still exhibits significantly smaller CV values compared to the other orders. This suggests that the second-order difference provides a relatively more consistent measure of an instance's learning status across different samples, even when the CV values do not converge. The lower CV values of the second-order difference and the third-order difference throughout the training process support the use of a unified threshold $\delta$ to determine the *mastered* instances. This property simplifies the implementation and management of the mastered criterion in the IES method, as it allows for a more consistent approach to identifying *mastered* instances across the entire dataset. Using the second-order difference ($N = 2$) as the mastered criterion achieves good performance in most cases, as shown in Figure 5. $N = 2$ outperformed other configurations (including $N = 3$) in most scenarios. Given the satisfactory performance of $N = 2$, the potential benefits of exploring higher-order differences ($N > 3$) may be limited. The additional computational complexity introduced by higher-order differences may not yield significant improvements in the effectiveness of the IES method.

These experimental results provide evidence for the effectiveness of using the second-order difference as the *mastered* criterion in the IES method, enabling a more efficient and generalizable approach to instance-dependent early stopping.

# D    COMPARE WITH VARYING METHODS AND CRITERIA

To evaluate the effectiveness of the proposed IES method and its different criteria, we conducted experiments comparing IES with other sample selection methods under various hyperparameter settings. Figure 5 presents the results of these experiments on CIFAR-10 and CIFAR-100 datasets. It is worth noting that the hyperparameters were fine-tuned to manually set the methods and criteria to have similar total backpropagation sample savings rates, making the methods comparable.

### D.0.1    EXPERIMENT: IES WITH DIFFERENT CRITERIA, HYPERPARAMETERS AND COMPARISON METHODS

- **Setup**:
  - Models: ResNet-18 for CIFAR-10, ResNet-34 for CIFAR-100
  - Optimizers: SGD with momentum and exponential decay, the initial learning rate is set to 0.1, and the gamma parameter is set to 0.96
  - Training Epoch: 200 for CIFAR
  - Batch Size: 64 for CIFAR

  - Comparison Methods (CIFAR-10 and CIFAR-100):
    * *Random Remove*: Randomly excludes a fixed proportion of samples from backpropagation in each training epoch. Removal rates: 10%, 20%, 30%, 40%, 50%.
    * *Small Loss & Reweight* (Qin et al., 2023): Randomly removes samples with smaller loss values and amplifies the gradients of the remaining small-loss samples. To focus on the core idea of the method and ensure a simple and direct comparison with the proposed IES method, we removed the annealing and other additional operations from the original implementation. This modification allows us to evaluate the effectiveness of removing small-loss samples and amplifying their gradients in isolation, providing a clearer understanding of the differences between the two methods. Removal ratios: 10% - 50%. A comparison of the wall-time between IES method and InfoBatch method is provided in Figure 7.

- **Results**:
  - IES with $N = 2$ (2nd order difference) outperforms other criteria and sample selection methods in most cases, achieving a good balance between computational efficiency and model performance.
  - The performance of IES is relatively stable across a wide range of $\delta$ values for each criterion.
  - *Random Remove* significantly reduces model performance, confirming the effectiveness of the IES method in selecting not-yet *mastered* samples.
  - *Small Loss & Rescale* improves results compared to *Random Remove* but still falls behind IES.

Figure 5 visualizes the results of these experiments, comparing the test accuracy of different methods and criteria under varying Total Excluded Samples ratios.

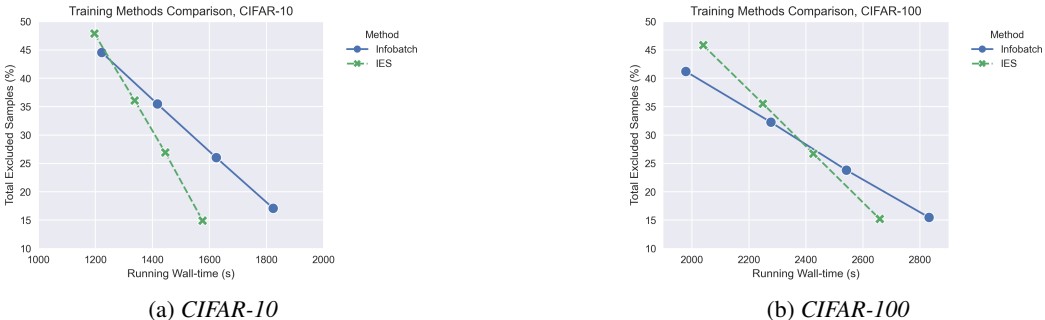

|         (a) *CIFAR-10*          |          (b) *CIFAR-100*          |

Figure 7: Comparison of the wall-time between IES method and InfoBatch method

# E    HIGH-LEVEL TASKS

To further validate the general applicability of our perspective and method, we provide a comprehensive evaluation of our proposed Instance-dependent Early Stopping (IES) method across two distinct but equally important high-level vision tasks: object detection and image segmentation, providing a broader perspective on its potential applications in the field of computer vision. Our experimental approach centered on integrating our proposed IES method into established baseline models for each task. Here's a detailed look at the experimental setup and result for each task:

We selected Faster R-CNN (Ren, 2015) as our baseline for object detection. Faster R-CNN is a two-stage detector that has shown remarkable performance in accurately identifying and localizing multiple objects within an image. For this experiment, we utilized the PASCAL VOC2007 (Everingham et al., a) dataset, and we implemented VGG-16 (Simonyan & Zisserman, 2014) as the backbone network for feature extraction. Both the baseline method and the IES method were run for 30 epochs. We evaluate the best mAP value of the trained model and report the proportion of back-propagation mini-batches saved by the IES method.

| Object Detection | | |
|---|---|---|
| | mAP (%) | Mini-Batch Saved (%) |
| Baseline | 70.2 ± 0.2 | \ |
| InfoBatch (Qin et al., 2023) | 69.9 ± 0.2 | 18.7 |
| IES (Ours) | 70.2 ± 0.1 | 20.0 |

For the task of image segmentation, we chose DeepLab v3 (Chen, 2017) as our baseline. DeepLab v3 is a state-of-the-art model for semantic segmentation, allowing the model to capture multi-scale contextual information effectively. We employed the PASCAL VOC2012 (Everingham et al., b) dataset for this experiment, and we used ResNet-50 (He et al., 2016) as the backbone network. Both the baseline method and the IES method were run for 50 epochs. We evaluate the best mIoU value of the trained model and report the proportion of back-propagation mini-batches saved by the IES method.

| Image Segmentation | | |
|---|---|---|
| | mIoU (%) | Mini-Batch Saved (%) |
| Baseline | 76.2 ± 0.2 | \ |
| InfoBatch (Qin et al., 2023) | 76.0 ± 0.3 | 12.0 |
| IES (Ours) | 76.1 ± 0.2 | 14.0 |

# F    MORE BASELINE METHODS

We further compare the IES method with several other data efficient methods, including:

1. The conventional early stopping method.

2. The importance sampling method (Jiang et al., 2019).

3. Curriculum learning methods (Zhou et al., 2020; Wang et al., 2024b).

To evaluate the applicability of the IES method in scenarios where efficiency is the primary objective, we conducted comparisons using the same training parameters as the IES method (detailed in Section D). To further demonstrate the ability of these methods to accelerate training while tolerating a certain degree of model performance degradation, we reduced the total training epochs by half to 100 and set the target computational speedup to approximately 2.0 and 3.0 times. Under higher speedup ratios, we evaluate the loss on the full training set at five-epoch intervals to reduce the computational overhead of loss evaluation, thereby enabling more efficient training. The comparison is made based on the test accuracy achieved by each method's trained model.

Table 8: Comparison of IES and other data efficiency methods. (3 runs, mean±std)

| Computation Speedup | Methods | CIFAR-10 | CIFAR-100 |
|---|---|---|---|
| 1.0× | Baseline (No Removal) | 94.3%±0.3% | 77.0%±0.4% |
| ∼ 2.0× | Conventional Early Stopping | 90.4%±0.5% | 68.7%±0.5% |
| | SB (Jiang et al., 2019) | 93.0%±0.1% | 70.6%±0.5% |
| | DIHCL (Zhou et al., 2020) | 93.4%±0.2% | 74.3%±0.2% |
| | EfficientTrain (Wang et al., 2024b) | 91.5%±0.2% | **75.0%±0.1%** |
| | IES (Ours) | **93.7%±0.4%** | 74.9%±0.5% |
| ∼ 3.0× | Conventional Early Stopping | 88.1%±0.3% | 63.9%±1.0% |
| | SB (Jiang et al., 2019) | 91.1%±0.5% | 65.8%±0.3% |
| | DIHCL (Zhou et al., 2020) | 92.7%±0.1% | 72.6%±0.1% |
| | EfficientTrain (Wang et al., 2024b) | 92.5%±0.2% | 70.6%±0.7% |
| | IES (Ours) | **93.2%±0.1%** | **73.0%±0.5%** |

As shown in Table 8, these comparisons further demonstrate that IES, while not specifically designed for scenarios where efficient training is the primary objective, still performs effectively in accelerating model training while maintaining model performance. This can be attributed to its adaptively identifying and excluding *mastered* samples during the training process.

# G  LABEL NOISE

An analysis of learning with noisy labels (Yao et al., 2020; 2021; Wei et al., 2022; Yuan et al., 2023; Lin et al., 2023; 2024d; Wu et al., 2024) is crucial to evaluate the robustness and practicality of our proposed IES method. To address this, we attempt to discuss this issue under Typical Learning with Noisy Label scenarios and Epoch-wise Double Descent scenarios, respectively.

**Typical Learning with Noisy Labels.**   We validate the performance of the IES method and the baseline method (without removal) under typical learning with noisy labels settings, specifically, on the CIFAR-10/CIFAR-100 datasets with 20% and 40% symmetric and instance-dependent (Xia et al., 2020b) label noise.

Table 9: Performance comparison on CIFAR-10 dataset with different noise settings.

| Noise Ratio | Type | Best Accuracy [Early Stopping Epoch] | | Mini-batch Saved |
|---|---|---|---|---|
| | | Baseline | IES | |
| 20% | Symmetric | 87.81% [21] | 87.81% [21] | 0% |
| 40% | Symmetric | 81.29% [13] | 81.29% [13] | 0% |
| 20% | Instance | 87.09% [22] | 87.09% [22] | 0% |
| 40% | Instance | 83.49% [20] | 83.49% [20] | 0% |

Table 10: Performance comparison on CIFAR-100 dataset with different noise settings.

| Noise Ratio | Type | Best Accuracy [Early Stopping Epoch] | | Mini-batch Saved |
|---|---|---|---|---|
| | | Baseline | IES | |
| 20% | Symmetric | 55.39% [17] | 55.39% [17] | 0% |
| 40% | Symmetric | 43.87% [15] | 43.87% [15] | 0% |
| 20% | Instance | 57.30% [18] | 57.30% [18] | 0% |
| 40% | Instance | 47.67% [18] | 47.67% [18] | 0% |

The experimental results indicate that the IES method degenerates to the baseline method (without removal) across all tested label noise rates, noise types, and datasets. This suggests that during the training process, no training sample satisfies the master criterion before the model overfits to the noisy labels and its performance declines.

The core idea behind the IES method is that once a model has mastered a sample, it should stop training on that sample. However, when a certain proportion of label noise exists in the dataset, memorization of mislabeled samples may affect the model's ability to learn stable patterns, making it difficult for the model to truly master any samples before the early stopping point.

**Epoch-wise Double Descent.** Epoch-wise Double Descent refers to the phenomenon where, when the training samples contain a certain amount (usually low) of label noise, as training progresses, the model's generalization performance first rises, then falls, and then rises again, with the generalization performance after the second rise being superior to the first peak. In this label noise scenario, the model needs to prolong training to achieve better generalization performance compared to conventional early stopping. We validate the performance of the IES method and the baseline method (without removal) under typical Epoch-wise Double Descent settings, specifically, on the CIFAR-100 datasets with 10% symmetric and instance-dependent label noise.

Table 11: Performance comparison under Epoch-wise Double Descent settings on CIFAR-100.

| Noise Ratio | Type | Best Accuracy [Epoch] | | Mini-batch Saved |
|---|---|---|---|---|
| | | Baseline | IES | |
| 10% | Symmetric | 61.9% [190] | **62.0%** [191] | 14.2% |
| 10% | Instance | 58.9% [151] | **59.2%** [199] | 11.0% |

The experimental results show that the IES method can achieve lossless efficient training under the Epoch-wise Double Descent scenario. In the later stages of training, the model inevitably "well-learn" some instances due to the memorization effect. However, this does not affect the generalization performance of the final model (even slightly better).

This behavior can potentially be explained by the fact that although 'well-learned' instances may be forgotten as the model training overfits the mislabeled samples, the IES method allows these samples to adaptively re-include in training, thereby mitigating the negative impact of mislabeled samples. Furthermore, as shown in the Figure 4, the IES method can more targetedly reduce steepness in these sharp directions of the loss landscape, and therefore may be able to train a model with better generalization performance even in the presence of label noise.

Consequently, in the typical scenarios of learning with noisy labels and scenarios of Epoch-wise Double Descent, the IES method appears to have no negative impact on model performance compared to the baseline.

## H  CATASTROPHIC FORGETTING

We define "early removed examples" as the first 5% of samples that are removed. We conducted experiments in a typical IES training environment with CIFAR-10, ResNet18, and SGD optimizer, which saves approximately 43% of the backpropagation samples in total 200 training epoch.

We tracked the average training loss and accuracy of these "early removed examples" during the training process and compared them with the corresponding values of the entire training set. The experimental results are as follows:

The results demonstrate that the "early removed examples" are well learned (even better) by the model, and their training accuracy and loss are on par with other samples in the end of training. This implies that the model isn't catastrophically forgetting these "early removed examples".

Furthermore, we investigated the reasons why our IES method does not lead to catastrophic forgetting. Notably, the IES is a reversible method, which means that the removed samples have a chance to re-include in the training process if their second-order loss difference exceeds the threshold. Therefore, we tracked the average number of times the "early removed examples" were re-included in the training process, as shown in the following table:

Considering that our method allows these "early removed examples" to re-include in training for an average of about 13 times, with the most frequently replaced samples experiencing 26 training

Table 12: Comparison of training loss and accuracy between full training set and early removed examples across different epochs.

| Epoch | Training Loss | | Training Accuracy | |
|---|---|---|---|---|
| | Full Set | Early Removed | Full Set | Early Removed |
| 50 | 0.120135 | 0.001003 | 95.96% | 100.00% |
| 100 | 0.001448 | 0.000920 | 99.99% | 99.98% |
| 150 | 0.000914 | 0.000806 | 100.00% | 100.00% |
| 200 | 0.000883 | 0.000833 | 100.00% | 100.00% |

Table 13: Statistics of sample re-inclusion during training.

| Metric | Value |
|---|---|
| Average Times Re-included | 13.14 |
| Maximum Times Re-included | 26.00 |

replays, we propose that this adaptive dynamic training mechanism contributes to the IES method's ability to effectively prevent "early removed examples" from being catastrophically forgotten during model training.

## I   FAIRNESS

We conducted a preliminary assessment of the fairness of training using the IES method in sensitive environments. We utilized the CelebA face dataset as an adversarial dataset to investigate whether the IES method would introduce new biases during training when using male as the sensitive attribute and attractiveness as the target label, thereby affecting the model's fairness.

We compared the baseline method (without sample removal) and the IES method on the ResNet-18 model for the attractiveness classification task, evaluating the accuracy, recall (True Positive Rate), and Demographic Parity Difference (DPD) metrics on the male and female validation subsets. The results are as follows:

Table 14: Fairness evaluation on CelebA dataset using gender as the sensitive attribute. Metrics include overall accuracy, gender-specific accuracy and recall rates, and Demographic Parity Difference (DPD). Lower DPD indicates better fairness. Best results are shown in **bold**.

| Method | Overall | Male | | Female | | DPD |
|---|---|---|---|---|---|---|
| | Acc. | Acc. | Recall | Acc. | Recall | |
| Baseline | **82.5** | **83.8** | **68.2** | 81.6 | **90.6** | 0.4613 |
| IES (Ours) | 82.4 | 83.4 | 58.9 | **81.8** | 87.0 | **0.4544** |

From the Demographic Parity Difference (DPD) metric, which evaluates fairness (the closer to 0, the better), the IES method is slightly lower than the baseline method (0.4544 vs 0.4613), indicating that its prediction results have slightly less disparity between the two gender subsets.

These results provide a preliminary indication that the IES method may introduce or amplify certain biases to some extent, negatively impacting the classification performance for different population subsets. However, since IES allows excluded samples to adaptively re-participate in training, the overall fairness is slightly improved.

