# OpenReview forum: "Instance-dependent Early Stopping"
_ICLR.cc/2025/Conference — ICLR 2025 Spotlight_

### Official Review · Reviewer_fBn3 · 2024-10-28

**Soundness:** 4
**Presentation:** 4
**Contribution:** 3
**Rating:** 8
**Confidence:** 3

**Summary:**

This paper introduces instance-dependent early stopping (IES), which refines the idea of early stopping from the entire training dataset to the instance level. The method uses second-order loss information to determine whether the data sample was mastered or not. The authors showed empirically that using second order information gives better results compared to zeroth, first and third order information (Figure 5). Finally, the authors provide a range of experiments showing great results of their methodology. Importantly, and as mentioned in Sec. 5, the analysis is focused on dataset-level performance and class-level performance (performance on underrepresented groups) is not taken under consideration in this work.

**Strengths:**

This paper was a very good read. The idea is simple, intuitive and provides a very elegant in-between to curriculum learning, dynamic data pruning and early stopping. The flow of the paper is very good, and the literature review feels very strong to me (only with some minor issues in relation to dynamic data pruning that I mention later). The experiments are well thought through and feel reasonably extensive to me. Finally, the results are great, which is a very good sign for a methodology paper. I think that it’s a strong paper that should be accepted and will be of interest to the research community, especially due to connection to various research fields mentioned in Section 2, and clear improvements to very popular algorithm (early stopping). I also really appreciate that the authors have explicitly pointed out their lack of studies on underrepresented groups in Limitations, clearly opening path for future research.

**Weaknesses:**

# Reasoning behind second-order information
On one hand I do think that information in Figure 5 is enough to validate using second-order loss information instead of higher- or lower-order information. On the other hand, I cannot agree with a statement that “the choice of using the second-order difference as the removal criterion for IES has been **extensively** validated through experiments” (L537-538). This statement feels too strong in my opinion. For that to be the case I would expect to see third-order information in Figures 3 and 6.

# Distinction from dynamic data pruning
(L147-L151) I am not completely sold on this argument. IES feels to me like a variant of dynamic data pruning. Because of that I also do not understand why the authors did not include training times in Figure 5, especially as it’s the only experiment where they compare their method with dynamic data pruning methodologies. I also don’t understand why the authors used only one work for comparison (I don’t have much knowledge in dynamic data pruning though so I base this comment on the brief analysis of the relevant literature provided by the authors).

# Potential improvements in clarity (minor)
1. (Eq. 2) I assume that the formula comes from Taylor approximation, but this is something that the authors should explicitly state rather than leave for the reader to deduce.
2. It is not clear why different optimizers were used for different architectures. This should be either explained or a reference should be provided (unless it is somewhere in the appendix and I missed it).
3. Typos in L137, L328, L368, L423, and L1004
4. (L295-L296)  “As shown in Figure 1, our method can reduce the number of training instances in backpropagation by over 40%” - please specify that it’s only in the later stages of the training. Otherwise it’s too ambiguous and could be considered as overselling results.

# Lack of fully extensive experiments

It would be interesting to see the comparison for different speedups (e.g., Table 4). Maybe IES becomes more effective, in relation to other methods, as we increase the speedup.

**Questions:**

# Clarification
1. Why did the authors not use FLOPs in their time analysis? Aren’t FLOPs more reliable for this setting? Also, why are there no precise times stated? It feels to me that it’s a very simple thing to do that would give a better perspective on speedups.
2. (L425-L426) Did the authors use IES in fine-tuning as well? Why or why not?
3. When authors compare the results of their IES with the baseline (No removal), is early stopping incorporate into that baseline? I don't think that's something that has been explicitly stated, and it feels to me that if the baseline uses early stopping then it would be significantly clearer to simply call it traditional early stopping (TES) or something similar.
# Future direction
1. (In relation to Figure 2) It would be interesting to show what samples are forgotten (lost their mastered status after removing). I would expect that these are samples that are objectively hard (according to probe models), but subjectively easy for this particular model (at this particular initialization). If we could identify a subset of mastered samples that can be forgotten (unmastered), then we might be able to improve IES by not removing them (a follow-up question is whether this would actually improve the performance). I am curious if the authors thought about it or did any experiments in this direction.

---

> ### Author Response · Authors · 2024-11-21
> **Response to Reviewer fBn3 (Part 1)**
>
> We would like to thank the reviewer for your detailed review and insightful comments.
> Below, we address the questions you have raised.
>
> ---
> >**Q1: Reasoning behind second-order information**
>
> A1: We agree with your point that the statement "the choice of using the second-order difference as the removal criterion for IES has been extensively validated through experiments" (L537-538 in the original manuscript) is too strong. We have removed the word "extensively" from this statement in the revised manuscript.
> Furthermore, as per your suggestion, we have added third-order information to Figures 3 and 6 in the revised manuscript. The updated figures demonstrate that the coefficient of variation (CV) for third-order and second-order loss values are close, reflecting a similar degree of dispersion. Considering the lower computational and storage costs of second-order differences, along with their slightly better experimental results, we opted to use second-order differences as the criterion for IES in the manuscript.
>
> ---
> > **Q2: Relationship to Dynamic Data Pruning**
>
> A2: When exploring the commonalities between the two strategy, we agree that our IES method could be viewed as a variant of dynamic data pruning that adaptively adjusts the removal rate based on the model's current learning status on training data. However, we humbly emphasize that IES differs from traditional data pruning methods in its ability to adaptively remove samples as the model training progresses, rather than following a predefined removal rate.
>
> Regarding the comparison with dynamic data pruning methods, we initially focused on a SOTA method to demonstrate the performance of our proposed selection criterion. To further clarify, we have conducted additional experiments comparing IES with another dynamic data pruning criterion ϵ-greedy [C1]. The results are as follows:
>
> | Method                 | CIFAR-10 Accuracy | Saved Ratio (%) |
> |------------------------|-------------------|-----------------|
> | IES (Ours)             | **95.0%**           | **43.7%**           |
> | ϵ-greedy | 94.8%             | 40.0%           |
>
> | Method                 | CIFAR-100 Accuracy | Saved Ratio (%) |
> |------------------------|--------------------|-----------------|
> | IES (Ours)             | **77.3%**                  | **45.8%**           |
> | ϵ-greedy | 75.5%                  | 40.0%           |
>
> As demonstrated, our proposed selection criterion consistently maintains higher performance compared to the dynamic data pruning method.
>
> Furthermore, in response to your constructive suggestion, we have included a comparison of the wall-time between IES and the dynamic data pruning method (InfoBatch) from our manuscript in Appendix D. The brief results are presented below:
>
> CIFAR-10:
>
>  | Method    | Saved Ratio (%) | Wall-time (s) |
> |-----------|-------------------|---------------|
>  | Infobatch | 44.56             | 1222.10       |
>  | Infobatch | 35.47             | 1417.48       |
>  | Infobatch | 26.03             | 1623.61       |
>  | Infobatch | 17.11             | 1823.21       |
>  | IES       | 47.90             | 1196.02       |
>  | IES       | 36.08             | 1337.78       |
>  | IES       | 26.96             | 1444.49       |
>  | IES       | 14.92             | 1575.90       |
>
>  CIFAR-100:
>
> | Method    | Saved Ratio (%) | Wall-time (s) |
> |-----------|-------------------|---------------|
> | Infobatch | 41.20             | 1978.82       |
> | Infobatch | 32.27             | 2276.72       |
> | Infobatch | 23.83             | 2541.40       |
> | Infobatch | 15.50             | 2831.13       |
> | IES       | 45.84             | 2039.22       |
> | IES       | 35.54             | 2248.92       |
> | IES       | 26.75             | 2425.57       |
> | IES       | 15.23             | 2658.29       |
>
> The results show that, at similar removal ratios, IES saves more wall-time on CIFAR-10, while achieving comparable results on CIFAR-100. This difference may be attributed to the shorter pipeline of our proposed method.

---

> ### Author Response · Authors · 2024-11-21
> **Response to Reviewer fBn3 (Part 2)**
>
> ---
> >**Q3: Potential improvements in clarity (minor)**
>
> A3: Thank you for your thorough review and valuable suggestions. We have addressed each of these points in the revised manuscript to improve clarity.
>
> - Eq. 2 defines the second-order difference of the loss function $L_i(w^{(t)})$ for a given sample $i$ at the current training epoch $t$: $ \Delta^2 L_i(w^{(t)}) = [L_i(w^{(t)}) - L_i(w^{(t-1)})] - [L_i(w^{(t-1)}) - L_i(w^{(t-2)})] = L_i(w^{(t)}) - 2L_i(w^{(t-1)}) + L_i(w^{(t-2)}). $
>
> - The use of different optimizers and architectures is intended to demonstrate the applicability of the proposed IES method under various settings.
>
> - We have corrected the typos in L137, L328, L368, L423, and L1004 in the original manuscript.
>
> - Concerning the statement in L295-L296, "As shown in Figure 1, our method can reduce the number of training instances in backpropagation by over 40%", the reference to Figure 1 was incorrect. In the revised manuscript, we have updated the reference to Figure 4 and rephrased the statement to be more precise: ''As shown in Figure 4, our method reduces the number of training instances in backpropagation by approximately 40%''
>
> ---
> >**Q4: Extensive experiments in Speedups**
>
> A4: Thank you for the suggestion. We have included an additional scenario where the computation speedup is increased to 3x, and the results are now in Appendix F. The brief results are shown below:
>
> | Computation Speedup | Methods                                    | CIFAR-10       | CIFAR-100      |
> |---------------------|--------------------------------------------|----------------|----------------|
> | 1.0×                | Baseline (No Removal)                      | 94.3%±0.3%     | 77.0%±0.4%     |
> | ~3.0×               | Conventional Early Stopping                | 88.1%±0.3%     | 63.9%±1.0%     |
> |                     | SB                  | 91.1%±0.5%     | 65.8%±0.3%     |
> |                     | DIHCL              | 92.7%±0.1%     | 72.6%±0.1%     |
> |                     | EfficientTrain       | 92.5%±0.2%     | 70.6%±0.7%     |
> |                     | IES (Ours)                                 | **93.2%±0.1%** | **73.0%±0.5%** |
>
> As shown in the table, the experimental results jive with your intuition: IES demonstrates increased effectiveness compared to other methods as we further increase the speedup.
>
> ---
> **We now turn to address the questions raised by the reviewer.**
>
> ---
> >**Q5: Clarification**
>
> - We agree that FLOPs can be a reliable metric for comparing computational costs. However, in our experiments, we found that the actual running time ratio better reflects the real-world computational overhead, as many of the computations are performed on the CPU (often with limited threads) in serial execution.
> Furthermore, although we ensured that each comparative method was run with exclusive access on identical nodes, we cannot guarantee that all tasks were run on nodes with the same configuration. Therefore, for fairness, we reported the average ratio of running times for a given task.
>
> - We did not use the IES method during fine-tuning because our goal was to simply evaluate the transferability of the trained models.
>
> - The baseline (No removal) does not incorporate conventional early stopping. We compared conventional early stopping in the speedup experiments presented in Table 4.
>
> We hope these explanations address your concerns and provide a clearer understanding of our experimental setup and choices.

---

> ### Author Response · Authors · 2024-11-21
> **Response to Reviewer fBn3 (Part 3)**
>
> ---
> >**Q6: Future direction**
>
> A6: You raised a very interesting point. To address this issue, we tracked 2,500 samples that were excluded earliest during the training of the CIFAR-10 dataset. Throughout the entire training process, we monitored the number of transitions these samples underwent from mastered to unmastered states, during which the model's performance changed significantly. The results showed that these samples had an average of 13.14 unmastered-to-mastered transitions, with a maximum of 26 transitions and a minimum of 0 transitions. Additionally, we tracked the loss changes for this subset of samples:
>
> | Epoch | Full Training Set Loss | Early Removed Set Loss |
> |-------|------------------------|------------------------|
> | 50    | 0.120135              | 0.001003              |
> | 100   | 0.001448              | 0.000920              |
> | 150   | 0.000914              | 0.000806              |
> | 200   | 0.000883              | 0.000833              |
>
> The results indicate that although the frequencies at which these samples were re-included in training varied, with some samples never being re-included in training, the model's learning performance on these samples consistently approached (or surpassed) its performance on the entire training set. This can be attributed to the adaptiveness of the IES method, where the mastered state of samples always depends on the current stage of model learning and changes dynamically. Therefore, we tend to believe that, in contrast, predefining a static subset would not improve model learning and may instead lead to brute-force memorization of that subset.
>
> ---
>
> Thank you for your valuable time and effort in reviewing our manuscript. Your insightful suggestions have greatly contributed to improving our manuscript.
>
> Best regards,
>
> Authors
>
> ---
>
> Reference:
>
> [C1] Accelerating Deep Learning with Dynamic Data Pruning. arXiv.

---

> > ### Comment · Reviewer_fBn3 · 2024-11-25
> >
> > **Addressing the Authors' response**
> >
> > I thank the Authors for their extensive response. I love the idea of IES and how it elegantly merges various fields while improving on a very well known early stopping algorithm. However, I do not have enough expertise in regards to practical papers to be able to determine whether the authors have performed sufficient experiments for a spotlight at ICLR. As such, I will keep my score.
> >
> > **Follow up to Q6**
> >
> > Thank you very much for indulging my interests. I am curious to know where the number 2,500 came from. Was it hard coded or based on something? When you say *13.14 unmastered-to-mastered transitions*, do you mean that an average sample, within the 2,500 samples that are mastered the fastest, looses and regains it's mastered status 13.14 times? Does it mean that even though the data sample was removed from the training set, it managed to regain its mastered status? Also, what do you mean when you say re-included in training? From what I understood, once a sample becomes mastered it stops being of interest to IES (it appears Reviewer TP1Y has a similar understanding - *examples are completely dropped (too early) and **never revisited**.*). This is probably based on *once the model has mastered an instance, the training on
> > it should stop* and many other such statements (the fact that $D$ is missing $^{(T)}$ in L5 of Algorithm 1 is hard to catch). You should make it more clear whether the samples are or are not re-included in training based on their current mastered status.

---

> ### Author Response · Authors · 2024-11-26
>
> Dear Reviewer fBn3,
>
> We are genuinely grateful for your prompt response. We have made our best efforts to address the concerns raised by all reviewers and have included additional experiments to evaluate our method. We also sincerely apologize for any lack of clarity in our manuscript that may have led to misunderstandings about the algorithmic details of IES.
>
> In the IES method, excluded samples can be re-included in the training process if they no longer meet the "mastered" criterion. This is reflected in Algorithm 1, where lines 3-5 compute the "mastered" status of instances through forward propagation, and line 10 performs backpropagation only on the not-yet mastered instances ($\mathcal{D}^{(t)}$). The main text also mentions this property of IES (e.g., lines 268-269 and 416-419). We have denoted the full training set as ($\mathcal{D}^{(0)}$) and referred to it as the "full training set" in the algorithm to emphasize this point.
>
> Regarding the 2,500 samples mentioned in our response to Q6, this number represents the top 5% earliest excluded samples from the training set, which we selected as a case study to analyze the IES method. The analysis reveals that, on average, these samples undergo 13.14 transitions from non-mastered to mastered states throughout the training process. This means that although these samples were considered well-learned early on, they are re-included in the training process and are subsequently excluded again upon reaching the "mastered" criterion.
>
> Thank you again for your valuable comments.
>
> Best regards,
>
> Authors

---

### Official Review · Reviewer_1YJ5 · 2024-11-01

**Soundness:** 3
**Presentation:** 4
**Contribution:** 3
**Rating:** 8
**Confidence:** 4

**Summary:**

This paper proposes an early stopping mechanism called Instance-dependent Early Stopping (IES) that takes into account individual instances in the training set. It is based on a concept that once the model learns an instance, it should stop iterating more on that particular instance. They define an approach to measure when an individual instance has been “mastered” based on the second-order derivative of its difference in loss. They empirically evaluate different aspects of the proposed method and also show that their proposed algorithm speeds up the training process while maintaining the generalization performance of the model.

**Strengths:**

- The paper is well-written, presents an interesting and intuitive idea, and is well-positioned within the existing literature.
- The proposed early stopping criterion offers computational efficiency by avoiding back-propagation, leading to faster training and reduced computation without sacrificing generalization.
-  The empirical results are extensive and address critical questions related to the method's effectiveness.

**Weaknesses:**

- A crucial missing element in this study is an analysis of the impact of noisy samples within the training set.  It is likely that in the presence of noisy labels, correctly labeled samples would be quickly learned and excluded, leaving the model to train primarily on incorrect information. This could lead to detrimental effects, potentially causing the model to "forget" previously learned correct trends and suffer from significantly worse generalization performance compared to standard training. Since, in practice, the presence of noise in a dataset is often unknown, this limitation raises concerns about the practicality of the proposed method in real-world scenarios.  Specifically, the claim that "well-learned" instances become redundant may not hold true when noisy samples are present, as the model may need to revisit correctly labeled data to counteract the influence of noisy labels. While Figure 2 addresses the persistence of learning in a clean setting,  a corresponding analysis for noisy datasets is crucial to fully evaluate the robustness and practicality of the proposed method.
- The "mastered" criterion introduces two new hyperparameters: a threshold, delta, and a step count, N. Introducing additional hyperparameters necessitates further tuning and selection, which reduces the practicality of the method. While the authors demonstrate some stability with respect to these parameters, it remains unclear how to optimally select them for datasets significantly different in nature from CIFAR-10 and CIFAR-100. This raises concerns about the method's generalizability and ease of use across diverse datasets.

**Questions:**

1. In Figure 1, The rate of excluded samples differs significantly between CIFAR-10 and ImageNet (non-linear vs. linear). What factors might explain this observation?
Why is training stopped prematurely without "mastering" all the training samples? What is the rationale for not continuing training beyond ~150 epochs?

2. Does the reported training loss in Figure 4 represent the loss calculated over the entire dataset (including mastered/dropped samples) or only on the remaining samples in each training epoch?

3. Figure 4 indicates that IES typically exhibits higher gradient norms. Could this lead to instability in certain settings due to larger steps and jumps during iterations? If so, under what conditions might this occur, and how can such instability be mitigated?

---

> ### Author Response · Authors · 2024-11-21
> **Response to Reviewer 1YJ5 (Part 1)**
>
> We would like to thank the reviewer for the detailed review and constructive comments.
> We sincerely appreciate your time and effort in reviewing our manuscript.
> Below, we address the questions you have raised.
>
> ---
> >**Q1: Label Noise**
>
> A1: Thank you for your insightful and constructive suggestion. We agree that an analysis of learning with noisy labels is crucial to evaluate the robustness and practicality of our proposed IES method.
> To address this, we attempt to discuss this issue under Typical Learning with Noisy Label scenarios and Epoch-wise Double Descent scenarios, respectively.
>
> **Typical Learning with Noisy Labels**
>
> We validate the performance of the IES method and the baseline method (without removal) under typical learning with noisy labels settings, specifically, on the CIFAR-10/CIFAR-100 datasets with 20% and 40% symmetric and instance-dependent label noise.
>
> **CIFAR-10**
>
> | Noise Ratio | Noise Type | Baseline Best Accuracy - [Early Stopping Epoch] | IES Best Accuracy - [Early Stopping Epoch] | Mini-batch Saved (%) |
> | --- |------------|-------------------------------------------------| --- | --- |
> | 20% | Sym.       | 87.81% - [21]                                   | 87.81% - [21] | 0 |
> | 40% | Sym.       | 81.29% - [13]                                   | 81.29% - [13] | 0 |
> | 20% | Inst.      | 87.09% - [22]                                   | 87.09% - [22] | 0 |
> | 40% | Inst.      | 83.49% - [20]                                   | 83.49% - [20] | 0 |
>
> **CIFAR-100**
>
> | Noise Ratio | Noise Type | Baseline Best Accuracy - [Early Stopping Epoch] | IES Best Accuracy - [Early Stopping Epoch] | Mini-batch Saved (%) |
> | --- |------------|-------------------------------------------------| --- | --- |
> | 20% | Sym.       | 55.39% - [17]                                   | 55.39% - [17] | 0 |
> | 40% | Sym.       | 43.87% - [15]                                   | 43.87% - [15] | 0 |
> | 20% | Inst.      | 57.30% - [18]                                   | 57.30% - [18] | 0 |
> | 40% | Inst.      | 47.67% - [18]                                   | 47.67% - [18] | 0 |
>
> The experimental results indicate that the IES method degenerates to the baseline method (without removal) across all tested label noise rates, noise types, and datasets. This suggests that during the training process, no training sample satisfies the master criterion before the model overfits to the noisy labels and its performance declines.
> The core idea behind the IES method is that once the model has mastered an instance, the training on it should stop.
> However, when a certain proportion of label noise exists in the dataset, memorization of mislabeled samples may affect the model's ability to learn stable patterns, making it difficult for the model to truly master any samples before the early stopping point.
>
>
> **Epoch-wise Double Descent**
>
> Epoch-wise Double Descent refers to the phenomenon where, when the training samples contain a certain amount (usually low) of label noise, as training progresses, the model's generalization performance first rises, then falls, and then rises again, with the generalization performance after the second rise being superior to the first peak. In this label noise scenario, the model needs to prolong training to achieve better generalization performance compared to conventional early stopping. We validate the performance of the IES method and the baseline method (without removal) under typical Epoch-wise Double Descent settings, specifically, on the CIFAR-100 datasets with 10% symmetric and instance-dependent label noise.
>
> | Noise Ratio | Noise Type | Baseline Best Accuracy - [Epoch] | IES Best Accuracy - [Epoch] | Mini-batch Saved (%) |
> | --- |------------|-------------------------------------------------| --- | --- |
> | 10% | Sym.       | 61.9% - [190]                                   | **62.0%** - [191] | 14.2% |
> | 10% | Inst.      | 58.9% - [151]                                   | **59.2%** - [199] | 11.0% |

---

> ### Author Response · Authors · 2024-11-21
> **Response to Reviewer 1YJ5 (Part 2)**
>
> The experimental results show that the IES method can achieve lossless efficient training under the Epoch-wise Double Descent scenario. In the later stages of training, the model inevitably "well-learn" some instances due to the memorization effect. However, this does not affect the generalization performance of the final model (even slightly better).
> This behavior can potentially be explained by the fact that although ''well-learned'' instances may be forgotten as the model training overfits the mislabeled samples, the IES method allows these samples to adaptively re-include in training, thereby mitigating the negative impact of mislabeled samples.
> Consequently, in the typical scenarios of learning with noisy labels and scenarios of Epoch-wise Double Descent, the IES method appears to have no negative impact on model performance compared to the baseline.
>
> This analysis is included in Appendix G of the revised manuscript.
>
> ---
> >**Q2: Hyperparameters**
>
> A2: The "mastered" criterion introduces only two hyperparameters: a threshold $\delta$ and an order N (in our experiments, we select N=2 for the IES method). While acknowledging these hyperparameters, our sensitivity analysis shows that model performance remains stable across a wide range of $\delta$ values, suggesting that precise tuning is not required for these datasets. Importantly, when $\delta$ is set too low, IES simply degenerates to baseline training without any performance penalty. Conversely, when $\delta$ is set too high, IES's adaptive mechanism - which automatically re-includes "mastered" samples into training - helps preserve model performance.
>
> To further validate the generalizability of IES, we conducted additional experiments on four different datasets, keeping the code and hyperparameter settings identical to those used for CIFAR-10 (except for dataset-specific dataloader and model architecture). The results are as follows:
>
> | Datasets       | Baseline Accuracy (No Removal) | IES Accuracy (Ours) | IES Mini-Batch Saved (%) | InfoBatch Accuracy (Ours) | InfoBatch Mini-Batch Saved (%) |
> |----------------|--------------------------------|-----------------|--------------------------|----------------------|--------------------------------|
> | SVHN           | 96.6%                          | **96.7%**          | **40.8%**                    | 96.5%                | 38.3%                          |
> | STL-10         | **72.1%**                      | 71.8%           | **45.0%**                    | 71.0%                | 42.5%                          |
> | Oxford Flowers | 59.8%                          | **59.9%**           | **20.1%**                    | 58.6%                | 17.8%                          |
> | Stanford Cars  | **61.0%**                      | **61.0%**           | **10.7%**                    | 58.9%                 | 9.5%                           |
>
> Besides comparing with the baseline, we also compared with the InfoBatch method at corresponding removal rates. As shown, without hyperparameter tuning, the IES method achieves superior performance in three training tasks; for the STL-10 task, it achieves a 45% reduction in backpropagation samples while only decreasing test accuracy by 0.2% compared with full data training.
> The results demonstrate that IES can efficiently and effectively train models on a variety of tasks without requiring targeted adjustment of hyperparameters.
>
> The robustness to parameter settings, along with the patterns observed across diverse datasets, suggests that the introduction of these hyperparameters does not significantly hinder the practicality, generalizability, or ease of use of the IES method.

---

> ### Author Response · Authors · 2024-11-21
> **Response to Reviewer 1YJ5 (Part 3)**
>
> ---
> **We now turn to address the questions raised by the reviewer.**
>
> ---
> >**Q3: Question about Figure 1**
>
> A3: The difference in the rate of excluded samples between different datasets can be attributed to the varying model capacities and task complexities.
> It's important to note that the "master" criterion in our work indicates that the model's performance improvement on a particular sample has become very small. This serves as an estimate of the sample's contribution to the current parameter updates of the model. Due to the correlations among samples, requiring the model to "master" all samples may be unnecessary and impractical. The model only needs to well-learn certain samples to achieve sufficiently good generalization performance.
>
> Moreover, as training progresses, the number of samples meeting the "master" criterion will eventually stabilize, with only minor fluctuations. This is because the contributions of samples to model parameter updates may be conflicting or not necessarily positive. In this situation, further extending the training will not significantly improve the generalization performance, as demonstrated by the following experiment on CIFAR-10:
>
> | Training Epoch | Test Accuracy at Epoch | Number of "Mastered" Samples |
> |----------------|------------------------|------------------------------|
> | 50             | 90.0%                  | 3,899                        |
> | 150            | 94.8%                  | 30,095                       |
> | 250            | 94.6%                  | 29,571                       |
> | 350            | 94.7%                  | 29,751                       |
>
> As shown in the table, the number of "mastered" samples remains relatively stable after 150 epochs. This suggests that the model has reached a point where further training does not significantly change the "mastered" status of the samples or improve the generalization performance.
> Consequently, as an early-stopping-based method aimed at computational efficiency, we choose 150 epochs as an appropriate end point for training in Figure 1 to evaluate the effectiveness and efficiency of our method under a limited computational budget.
>
> ---
> >**Q4: Training loss in Figure 4**
>
> A4: The training loss reported in Figure 4 represents the average loss computed over samples participating in training during each respective epoch.
>
> ---
> >**Q5: Higher gradient norms, present in Figure 4**
>
> A5: The IES method adaptively selects instances that are not yet well-learned by the model. These instances are likely to have larger loss values and, consequently, larger gradient norms. By focusing on these instances, IES enables the model to learn more efficiently and make more substantial progress in each iteration. Higher gradient norms in this context are a sign of effective instance selection and do not imply instability.
> We argue that the main reasons for IES's ability to maintain training stability are as follows:
> 1. IES employs second-order differences and moving averages as criteria for determining whether an instance has been mastered, rather than focusing on the instantaneous magnitude of gradients. Moreover, the adaptivity of the IES method allows for the re-include of instances whose "mastered" status has changed, in the event of potential training instability.
> 2. IES retains the vast majority of instances in the early stages of training and gradually reduces the number of instances in an adaptive manner as training progresses. This progressive instance selection strategy helps to maintain the smoothness of the optimization process.
> 3. We have already employed methods such as learning rate decay during training, which can mitigate the potential instability risks caused by large gradient updates. Furthermore, the further use of gradient clipping can further enhance training stability.
>
> ---
> We again thank the reviewer for the detailed and constructive feedback, which has helped us clarify key aspects of our method and improve the overall presentation of our work.
>
> Best regards,
>
> Authors

---

> ### Author Response · Authors · 2024-11-26
> **Comment for Reviewer 1YJ5**
>
> Dear Reviewer 1YJ5,
>
> We sincerely appreciate your detailed review and constructive comments on our manuscript. We are deeply grateful for your valuable time and effort.
>
> We have carefully considered your two main concerns and made the following efforts to address them in the revised version:
>
> 1. We conducted additional experiments under label noise scenarios. The results demonstrate that the IES method maintains performance on par with the baseline under typical learning with noisy labels and epoch-wise double descent settings, without introducing additional negative impact.
>
> 2. We validated IES hyperparameter generalization on four additional datasets, where IES performs better than competing methods without tuning hyperparameters, indicating that the hyperparameters do not significantly affect the method's practicality.
>
> We would greatly appreciate your further feedback on our revised version and our responses. If there are any unclear explanations, we are more than willing to provide further clarification.
>
> Best regards,
>
> Authors

---

> > ### Comment · Reviewer_1YJ5 · 2024-11-28
> > **Response to author replies**
> >
> > I would like to thank the authors for their detailed responses and for conducting additional experiments.
> >
> > I believe the experiments on label noise are strong additions to the paper's contributions. I would strongly suggest the authors revise their paper to put even more emphasize on these experiments.
> >
> > My main concern has been well-addressed with these experiments, so I have increased my original rating.

---

> > > ### Author Response · Authors · 2024-11-28
> > >
> > > Dear Reviewer 1YJ5,
> > >
> > > We are genuinely grateful for your prompt response. We agree the discussion of label noise makes our work more comprehensive, and we will revise the manuscript accordingly.
> > >
> > > Best regards,
> > >
> > > Authors

---

### Official Review · Reviewer_TP1Y · 2024-11-04

**Soundness:** 3
**Presentation:** 3
**Contribution:** 3
**Rating:** 6
**Confidence:** 4

**Summary:**

This work proposes a method, Instance-dependent Early stopping (IES), a variant of standard dataset-level early-stopping mechanism that in contrast uses an instance-level mastery threshold on second-order differences of the loss function to decide which examples to drop from the training (backward path). Calculation of this metric requires the forward path for the example to be dropped in the next epoch. This results in lossless acceleration in both nominal computational cost and real-world training time.

The empirical analysis presented in the paper quantifies these improvements in CIFAR-10/100 and ImageNet-1K. Multiple optimization algorithms and multiple learning rate schedules are analyzed separately. Results show nominal sample reduction of 10-55% and speedups of 20-40% and neutral accuracy. Further analysis shows that larger speedups are achievable if small suboptimality is allowed. Another empirical analysis is performed on the choice of hyperparameter (delta), establishing its effect on the error rate and training speedup.

**Strengths:**

- The paper is well motivated and easy to follow.

- Comparisons against the baseline and also early-stopping / data-efficiency  methods show strong SOTA in training acceleration for a few datasets.

- The speedups are not just in nominal sample complexity, but also in world-clock training time.


- The method only relies on the forward path and has an efficient implementation path in most ML frameworks.

**Weaknesses:**

- There is no theoretical analysis of the proposed method presented. There is also no rigorous suggestion on how one might attempt such analysis.

- There is no mention of catastrophic forgetting, which is a well-known and well-studied topic in non-uniform training (e.g. in RL and active learning). This setup is not as episodic as a full-blown RL domain, but there are clear similarities that would suggest catastrophic forgetting could occur if examples are completely dropped (too early) and never revisited. I suggest adding at least one experiment in which early dropped examples are evaluated at the end of training to measure an report any forgetting effects.

- The empirical test cases are very limited. SOTA results only on CIFAR-10/100 are not enough to establish the efficacy of the method on other settings. The study on the higher level (yet still vision) tasks is also somewhat limited and not against other data-efficiency methods. A comparison vs. other method can be added to the higher-level vision task study to improve the analysis.

- A very important point is raised in the limitations and not addressed in the paper: the fact that instance-based early stopping can introduce biases in the learning process that affect fairness or policy considerations. One could argue that without special care such side effects are bound to happen. To study this effect, one can build an adversarial dataset with easy examples from one protected group and harder ones from another group. Non-uniform training can result in shifted per-group precision/recall metrics compared to the baseline, which can be measured and reported.

**Questions:**

- How does the world-clock training time of the 1-order diff. method compare to the 2-order diff.? Calculating first-order differences might require less compute/memory even if the nominal complexity reduction is less than higher order differences.

- Figure 5, last row: training time reduction can also be added here + legend.

---

> ### Author Response · Authors · 2024-11-21
> **Response to Reviewer TP1Y (Part 1)**
>
> We would like to thank the reviewer for the detailed review and insightful comments. We sincerely appreciate your time and effort in reviewing our manuscript. Below, we address the questions you have raised.
>
> ---
> >**Q1: Theoretical Analysis**
>
> A1: As mentioned in the Limitations section, we acknowledge that a theoretical analysis is not present in this work.
> Our primary focus was to empirically validate the effectiveness of the proposed IES method in helping models achieve better performance in an efficient manner.
> The empirical validations indicate the practical value and rationality of our method. These empirical validations cover classification, segmentation, and detection tasks, using different combinations of backbone and optimizers, and on nine visual datasets.
>
> Additionally, we tracked various metrics during the training process, such as training loss, maximum eigenvalue of the Hessian matrix, and Sharpness-Aware Minimization (SAM) value. We utilized the Coefficient of Variation and experimental results to justify the selection of second-order differences as the criterion for sample removal. These measurements help explain our method's effectiveness.
>
> The second-order difference of the loss function, $\Delta^2 L(w)$, used in IES captures the change in the rate of change of the loss function values. This can be seen as an approximation of the second-order directional derivative of the loss function, which is related to the curvature of the loss landscape and the Hessian matrix [A1]. The second-order difference of loss provides an indication of the local curvature of the loss landscape. In regions where the curvature is high, the gradient is likely to change rapidly, indicating that the model is still actively learning from the corresponding samples. Conversely, in regions where the curvature is low, the gradient is likely to be more stable, suggesting that the model's learning from these samples is slowing down.
>
> Therefore, regarding potential directions for theoretical analysis, one could draw inspiration from previous works that have studied the use of gradient information for importance sampling. For example, Alain et al. [A2] analyzed importance sampling for SGD and proved that the variance-optimal sampling proposal $q^*(x)$ should be proportional to $p(x)||f(x)||_2$, where $p(x)$ is the data distribution and $f(x)$ is the gradient of the loss. Killamsetty et al. [A3] proposed selecting subsets to match the full gradient and proved convergence bounds under different conditions (Lipschitz continuity, smoothness, strong convexity).
>
> ---
> >**Q2: Catastrophic Forgetting**
>
> A2: Thank you for your insightful and constructive suggestion. We agree that it's crucial to check whether our method might cause the model to suffer from catastrophic forgetting. Taking your advice, we have looked into whether the "early removed examples" in the IES method would be catastrophically forgotten by the model.
>
> We define "early removed examples" as the first 5% of samples that are removed. We conducted experiments in a typical IES training environment with CIFAR-10, ResNet18, and SGD optimizer, which saves approximately 43% of the backpropagation samples in total 200 training epoch.
> We tracked the average training loss and accuracy of these "early removed examples" during the training process and compared them with the corresponding values of the entire training set. The experimental results are as follows:
>
> | Epoch   | Full Training Set Loss | Early Removed Set Loss |
> |---------|------------------------|------------------------|
> | 50      | 0.120135               | 0.001003               |
> | 100     | 0.001448               | 0.000920               |
> | 150     | 0.000914               | 0.000806               |
> | 200     | 0.000883               | 0.000833               |
>
> | Epoch   | Full Training Set Accuracy | Early Removed Set Accuracy |
> |---------|----------------------------|----------------------------|
> | 50      | 95.96%                     | 100.00%                    |
> | 100     | 99.99%                     | 99.98%                     |
> | 150     | 100.00%                    | 100.00%                     |
> | 200     | 100.00%                     | 100.00%                      |
>
> The results demonstrate that the "early removed examples" are well learned (even better) by the model, and their training accuracy and loss are on par with other samples in the end of training. This implies that the model isn't catastrophically forgetting these "early removed examples".

---

> ### Author Response · Authors · 2024-11-21
> **Response to Reviewer TP1Y (Part 2)**
>
> Furthermore, we investigated the reasons why our IES method does not lead to catastrophic forgetting. Notably, the IES is a reversible method, which means that the removed samples have a chance to re-include in the training process if their second-order loss difference exceeds the threshold. Therefore, we tracked the average number of times the "early removed examples" were re-included in the training process, as shown in the following table:
>
> | Metric                    | Value  |
> |---------------------------|--------|
> | Avg Times Re-included      | 13.14 |
> | Max Times Re-included      | 26     |
>
> Our proposed IES method allows these "early removed examples" to re-include in training for an average of about 13 times, with the most frequently re-include samples experiencing 26 training replays. We propose that this adaptive dynamic training mechanism contributes to the ability of IES method to effectively prevent "early removed examples" from being catastrophically forgotten during model training.
>
> This analysis is included in Appendix H of the revised manuscript.
>
> ---
> >**Q3: More Comparison**
>
> A3: We have conducted additional experiments on four diverse datasets: SVHN, STL-10, Oxford Flowers, and Stanford Cars. These datasets cover a range of image classification tasks, and the results are presented below:
>
> | Datasets       | Baseline Accuracy (No Removal) | IES Accuracy (Ours) | IES Mini-Batch Saved (%) | InfoBatch Accuracy (Ours) | InfoBatch Mini-Batch Saved (%) |
> |----------------|--------------------------------|-----------------|--------------------------|----------------------|--------------------------------|
> | SVHN           | 96.6%                          | **96.7%**          | **40.8%**                    | 96.5%                | 38.3%                          |
> | STL-10         | **72.1%**                      | 71.8%           | **45.0%**                    | 71.0%                | 42.5%                          |
> | Oxford Flowers | 59.8%                          | **59.9%**           | **20.1%**                    | 58.6%                | 17.8%                          |
> | Stanford Cars  | **61.0%**                      | **61.0%**           | **10.7%**                    | 58.9%                 | 9.5%                           |
>
> For all datasets, our proposed IES method consistently achieves comparable or even slightly better accuracy than the baseline while saving a significant rate of mini-batches during training. Compared to the Small Loss & Rescale criterion in InfoBatch method, IES consistently maintains higher accuracy while saving more mini-batches.
>
> Furthermore, as per your suggestion, we have included comparative experiments on high-level vision tasks, and the results have been added to Appendix E. The brief results are as follows:
>
> Object Detection, Faster R-CNN (3 runs, mean±std):
>
> | Method     | mAP (%)   | Mini-Batch Saved (%) |
> |------------|-----------|----------------------|
> | Baseline   | 70.2±0.2% | N/A                    |
> | InfoBatch  | 69.9±0.2%      | 18.7%                |
> | IES (Ours) | 70.2±0.1% | 20.0%                |
>
>  Segmentation, DeepLab v3 (3 runs, mean±std):
>
> | Method     | mIoU (%)   | Mini-Batch Saved (%) |
> |------------|------------|----------------------|
> | Baseline   | 76.2±0.2% | N/A                    |
> | InfoBatch  | 76.0±0.3%  | 12.0%                |
> | IES (Ours) | 76.1±0.2% | 14.0%                |
>
> These results demonstrate that our proposed "mastered" criterion can efficiently and effectively accelerate training compared to the Small Loss & Rescale criterion in InfoBatch method across both high-level vision tasks.

---

> ### Author Response · Authors · 2024-11-21
> **Response to Reviewer TP1Y (Part 3)**
>
> ---
> >**Q4: Fairness**
>
> A4: We appreciate your specific experimental suggestions, which indeed help us evaluate the limitations of the IES method that we pointed out in the Limitations section. Based on your recommendations, we conducted a preliminary assessment of the fairness of training using the IES method in sensitive environments.
>
> We utilized the CelebA face dataset as an adversarial dataset to investigate whether the IES method would introduce new biases during training when using **male** as the sensitive attribute and **attractiveness** as the target label, thereby affecting the model's fairness.
>
> We compared the baseline method (without sample removal) and the IES method on the ResNet-18 model for the attractiveness classification task, evaluating the accuracy, recall (True Positive Rate), and Demographic Parity Difference (DPD) metrics on the male and female validation subsets. The results are as follows:
>
> | Method  | Overall_Acc | Male_Acc | Male_Recall | Female_Acc | Female_Recall | DPD    |
> |--------|-------------|----------|-------------|------------|----------|--------|
> | Baseline | **82.5%**     | **83.8%**    | **68.2%**       | 81.6%      |**90.6%**    | 0.4613 |
> | IES (Ours)     | 82.4%       | 83.4%    | 58.9%       | **81.8%**      |87.0%    | **0.4544** |
>
> From the Demographic Parity Difference (DPD) metric, which evaluates fairness (the closer to 0, the better), the IES method is slightly lower than the baseline method (0.4544 vs 0.4613), indicating that its prediction results have slightly less disparity between the two gender subsets.
>
> These results provide a preliminary indication that the IES method may introduce or amplify certain biases to some extent, negatively impacting the classification performance for different population subsets. However, since IES allows excluded samples to adaptively re-include in training, the overall fairness is slightly improved.
>
> This analysis is included in Appendix I of the revised manuscript.
>
> ---
> **We now turn to address the questions raised by the reviewer.**
>
> ---
> >**Q5: World-clock Training Time of the 1-order Diff. Method Compare to the 2-order Diff.**
>
> A5: We compared the world-clock training time using the first-order difference criterion and the second-order difference criterion. Results are shown in the table below:
>
> | Criterion | Total Running Time (s) | Total Difference Calculation Time (s) | Best Accuracy | Saved Samples Ratio |
> |--------------------|------------------------|---------------------------------------|---------------|---------------------|
> | 2nd Diff.          | **1248.00**                | 5.16                                  | **95.0%**         | **43.7%**               |
> | 1st Diff.          | 1280.85                | **4.73**                                  | 94.8%         | 40.9%               |
>
> The results demonstrate that under similar backpropagation sample excluded ratios, the total running time of two methods is very close. Notably, compared to the first-order difference criterion, the second-order difference criterion only needs to store one additional epoch of loss values for the required calculations. The computational complexity also only increases from one arithmetic operation to three arithmetic operation (4.73s -> 5.16s), and the impact on the training time is very small (0.43s).
>
> ---
> >**Q6: Training Time Reduction in Figure 5 Last Row**
>
> A6: As per your suggestion, we have modified the last row of Figure 5 in the manuscript to include training time reduction.
>
> ---
>
> We extend our sincere thanks for your effort in reviewing our work and deeply appreciate your detailed comments towards improving our manuscript.
>
> Best regards,
>
> Authors
>
> ---
>
> Reference:
>
> [A1] Deep Learning via Hessian-Free Optimization. In ICML, 2010.
>
> [A2] Variance Reduction in SGD by Distributed Importance Sampling. In ICLR, 2016.
>
> [A3] GRAD-MATCH: Gradient Matching based Data Subset Selection for Efficient Deep Model Training. In ICML, 2021.

---

> ### Comment · Reviewer_TP1Y · 2024-11-25
> **Review on Updates**
>
> The updates to the paper include 3 crucial experiments that I believe were missing from the original submission: (1) a study of catastrophic forgetting in the context of the proposed method, (2) a study on the introduced unwanted bias, and (3) a larger experimental study against SOTA. The authors study some of the limitations of their work, which IMO opens up the opportunity to directly address them in follow up works. Other points and questions were also addressed by the updated submission. I have thus updated my assessment to reflect the changes made to the paper.

---

> ### Author Response · Authors · 2024-11-26
>
> Dear Reviewer TP1Y,
>
> We are genuinely grateful for your prompt response. Your feedback has helped us significantly improve our manuscript.
>
> Best regards,
>
> Authors

---

### Author Response · Authors · 2024-11-21
**Response to Reviewers**

Dear Reviewers, AC, and SAC,

We sincerely appreciate all the reviewers for their thoughtful feedback and efforts towards improving our manuscript. We tried our best to address all mentioned concerns/problems. Are there unclear explanations here? We could further clarify them.

Best regards,

Authors

---

### Meta-Review · Area_Chair_opNV · 2024-12-18

**Metareview:**

This paper introduces Instance-dependent Early Stopping (IES), a novel method for improving the efficiency of training machine learning models. Traditional early stopping halts training for the entire dataset when validation performance stops improving, but IES adapts this to stop training at the individual instance level. The method identifies an instance as "mastered" if the second-order differences in its loss value stabilize near zero, providing a consistent criterion for excluding instances from further backpropagation. By doing so, IES increases gradient norms, accelerates the reduction of training loss, and speeds up training. Experiments show that IES reduces backpropagation computations by 10%-50%, while maintaining or slightly improving model accuracy and transfer learning performance.

All the reviewers agree that this paper should be accepted. We encourage the authors to go through the reviewer comments and address it in the paper.

**Additional Comments On Reviewer Discussion:**

All the reviewers agree that this paper should be accepted. We encourage the authors to go through the reviewer comments and address it in the paper.

---

### Decision · Program_Chairs · 2025-01-22

Accept (Spotlight)